# SAD-Flower: Flow Matching for Safe, Admissible, and Dynamically Consistent Planning

**Tzu-Yuan Huang** [1]   **Armin Lederer** [2]   **Dai-Jie Wu** [3 4]   **Xiaobing Dai** [1]
**Sihua Zhang** [5]   **Hsiu-Chin Lin** [6]   **Shao-Hua Sun** [3 7]   **Stefan Sosnowski** [1]   **Sandra Hirche** [1 8 9]

## Abstract

Flow matching (FM) has shown promising results in data-driven planning. However, it inherently lacks formal guarantees for ensuring state and action constraints, whose satisfaction is a fundamental and crucial requirement for the safety and admissibility of planned trajectories on various systems. Moreover, existing FM planners do not ensure the dynamical consistency, which potentially renders trajectories inexecutable. We address these shortcomings by proposing SAD-Flower, a novel framework for generating **S**afe, **A**dmissible, and **D**ynamically consistent trajectories. Our approach relies on an augmentation of the flow with a virtual control input. Thereby, principled guidance can be derived using techniques from nonlinear control theory, providing formal guarantees for state constraints, action constraints, and dynamic consistency. Crucially, SAD-Flower operates without retraining, enabling test-time satisfaction of unseen constraints. Through extensive experiments across several tasks, we demonstrate that SAD-Flower outperforms various generative-model-based baselines in ensuring constraint satisfaction. Video and demos can be found at sadflowerplanning.github.io.

## 1. Introduction

Generative models have recently emerged as powerful tools for trajectory planning, with diffusion models (Ho et al., 2020) and flow matching (Lipman et al., 2023) enabling the generation of complex, long-horizon behaviors by directly learning from data. Unlike traditional data-driven planners that combine learned dynamics with optimization routines (Posa et al., 2014; Kalakrishnan et al., 2011), generative approaches avoid model exploitation—where optimizers produce trajectories that perform well under the model but fail in reality due to approximation errors (Talvitie, 2014; Ke et al., 2019). By training models to generate full trajectories that implicitly encode system dynamics, generative planners naturally capture multimodal (Huang et al., 2025), high-dimensional behaviors while mitigating compounding errors and supporting task compositionality. These advantages make generative approaches increasingly attractive for real-world planning and control tasks.

Despite these advantages, a critical limitation of generative model planners lies in their inability to guarantee constraint satisfaction – specifically, state and action constraints. State constraints ensure safety, e.g., avoiding collisions, while action constraints guarantee admissibility, e.g., respecting torque or power limits, which makes these constraints essential in domains such as robotics (Craig, 2009). However, constraint satisfaction at individual time steps is insufficient: since trajectories are sequences of states (and actions), subsequent states in trajectories cannot be chosen independently (Kelly, 2017). For a planned trajectory to be physically realizable and executable on the system, it must be dynamically consistent, i.e., planned states must evolve according to the system dynamics. However, existing generative planners offer no inherent mechanism to enforce such properties, especially under constraints unseen during training. These factors make constraint satisfaction particularly challenging at test time and motivate the need for formally grounded methods that can reliably ensure safety, admissibility, and consistency in generated trajectories.

Several training-free methods have been proposed to address parts of this problem, including guidance-based sampling (Yuan et al., 2023), e.g., Classifier Guidance (Dhariwal & Nichol, 2021), Decision Diffuser (Ajay et al., 2023), control-inspired guidance, e.g., CoBL-diffusion (Mizuta & Leung, 2024), and constraint-aware diffusion,

---

[1]TUM School of Computation, Information and Technology, Technical University of Munich, Munich, Germany. [2]National University of Singapore [3]National Taiwan University (NTU) [4]University of Utah [5]Beijing Institute of Technology [6]McGill University [7]NTU Artificial Intelligence Center of Research Excellence (NTU AI-CoRE) [8]Munich Institute of Robotics and Machine Intelligence (MIRMI) [9]Munich Data Science Institute (MDSI). Correspondence to: Tzu-Yuan Huang <tzu-yuan.huang@tum.de>.

*Proceedings of the 43rd International Conference on Machine Learning*, Seoul, South Korea. PMLR 306, 2026. Copyright 2026 by the author(s).

*Table 1.* **Generative model-based planner comparison.**

| Methods | State constraint | Action constraint | Dynamic consistency | Theoretical guarantee |
|---|---|---|---|---|
| Classifier Guidance (Dhariwal & Nichol, 2021) | ✓ | ✗ | ✗ | ✗ |
| CoBL-diffusion (Mizuta & Leung, 2024) | ✓ | ✗ | ✓ | ✗ |
| SafeDiffuser (Xiao et al., 2025) | ✓ | ✗ | ✗ | ✓ |
| Decision Diffuser (Ajay et al., 2023) | ✓ | ✗ | ✓ | ✗ |
| SAD-Flower (Ours) | ✓ | ✓ | ✓ | ✓ |

e.g., SafeDiffuser (Xiao et al., 2025) variants. While these approaches can adapt to test-time constraints, they remain fundamentally limited. As summarized in Table 1, most methods focus on state constraints, while action constraints and dynamic consistency are often neglected or addressed only implicitly. Moreover, they mostly rely on soft guidance, which lacks formal guarantees and can result in residual violations. This leaves a gap for methods that can provide theoretical guarantees for state and action constraints, and dynamic consistency at test time.

To address these challenges, we propose **SAD-Flower** – a novel control-augmented flow matching framework designed to generate **S**afe, **A**dmissible, and **D**ynamically consistent trajectories. Inspired by guidance-based approaches, SAD-Flower introduces a virtual control input into the generation process. This control-theoretic interpretation enables a principled design of test-time guidance signals to ensure strong guarantees. The foundation of the design lies in a novel reformulation of state and action constraints into Control Barrier Function (CBF) conditions (Ames et al., 2017) for the generation process, while dynamic consistency is transformed into a Control Lyapunov Function (CLF) condition (Sontag, 1983). These conditions are encoded as hard constraints in a constrained optimal control problem, solved efficiently using a quadratic program during sampling, allowing SAD-Flower to enforce novel constraints at test time without retraining. Unlike general constraint-projection approaches (Römer et al., 2025; Bouvier et al., 2025), which may distort the learned generative distribution, or control-guidance methods like CoBL (Mizuta & Leung, 2024), which offer no guarantees and operate only over action sequences, SAD-Flower directly enforces state and action constraints over full trajectories. By leveraging prescribed-time control (Song et al., 2017), we ensure satisfaction by the final step while preserving generation quality. These theoretical guarantees translate into empirical performance: SAD-Flower consistently satisfies all constraints across benchmarks and remains robust under increasingly strict test-time conditions. Moreover, we demonstrate its scalability to the dexterous grasping task, validating its practical applicability to high-dimensional robotic systems.

## 2. Related Work

**Diffusion and Flow-Based Generative Models for Planning.** Recent advances in generative modeling, including diffusion models (Sohl-Dickstein et al., 2015; Ho et al., 2020; Song et al., 2021) and flow-matching (Lipman et al., 2023; Albergo & Vanden-Eijnden, 2023), have shown remarkable performance across various domains such as image generation (Dhariwal & Nichol, 2021; Du et al., 2020; Hiranaka et al., 2025) and language modeling (Liu et al., 2023; Saharia et al., 2022). These generative approaches have also been successfully applied to data-driven planning, where the model learns to imitate expert behavior from datasets (Chen et al., 2024; Lai et al., 2024; Huang et al., 2024a). For example, some works generate entire state-action trajectories directly using one (Janner et al., 2022; Zheng et al., 2023) or two separate models (Zhou et al., 2025), while others predict high-level trajectories and rely on a downstream controller to compute low-level actions (Chi et al., 2024; Ajay et al., 2023; Ko et al., 2024; 2025). However, these generative model planners operate without mechanisms to ensure that generated trajectories respect real-world constraints. In particular, they lack formal guarantees for satisfying state and action constraints, as well as dynamic consistency.

**Constraint-Aware Generative Model Planner.** To address the issue of constraint satisfaction, several recent works have explored constraint-aware planning based on generative models. Guidance-based methods (Dhariwal & Nichol, 2021; Yuan et al., 2023; Kondo et al., 2024; Ma et al., 2025; Carvalho et al., 2023) incorporate constraints by injecting gradients of auxiliary cost functions into the sampling process. While this encourages constraint satisfaction, it provides only a soft inductive bias without formal guarantees. Classifier-free guidance (Ho & Salimans, 2022) leverages information, such as constraint violation (Ajay et al., 2023; Hung et al., 2025; Yeh et al., 2025) during training, enabling constraint-aware generation. However, these approaches require additional labeled data and have limited generalization to novel constraints. Similarly, DDAT (Bouvier et al., 2025) incorporates projection into the feasible set during training and inference but relies on strong assumptions, such as convexity of the constraint set. Post-processing approaches (Mazé & Ahmed, 2023; Giannone et al., 2023) attempt to enforce constraints by optimizing generated samples after denoising, but they are unaware of the learned data distribution and can produce samples that significantly drift from it.

**Control-Theoretic Enforcement in Generative Planning.** Several works have proposed control-theoretic techniques to enforce constraints. The works (Xiao et al., 2025; Botteghi et al., 2023; Dai et al., 2025) employ Control Barrier Functions (CBFs; Ames et al. 2017) to enforce state constraints

during the denoising process. However, these methods neglect action constraints and often produce trajectories that are not dynamically consistent and suffer from the local trap problem (Xiao et al., 2025). In CoBL (Mizuta & Leung, 2024), Control Lyapunov functions (CLFs; Sontag 1983) are used as a reward to promote goal-reaching behavior, but action constraints are not considered, and the guidance-based structure lacks formal guarantees. A constrained optimal control layer is integrated into the denoising process in (Römer et al., 2025) to enforce state and action constraints. Since it performs non-convex optimization throughout the entire sampling process, it exhibits a high computational cost and potentially steers samples prematurely before they reflect meaningful structure (Fan et al., 2025).

## 3. Problem Formulation

We formally define the trajectory planning problem with safety, admissibility, and dynamic consistency constraints as follows.

**System Model.** We consider a nonlinear dynamical system with state $s(k) \in \mathbb{R}^n$ and action $a(k) \in \mathbb{R}^m$ at time $k$, evolving as

$$s(k+1) = f(s(k), a(k)), \qquad (1)$$

where $f$ is the (possibly unknown) transition function. A trajectory is defined as a sequence of state-action pairs, $\tau = \{(s(0), a(0)), \ldots, (s(H-1), a(H-1))\}$. Given a dataset of expert trajectories $\mathcal{D} = \{\tau^{(n)}\}_{n=1}^N$, our goal is to learn planning new trajectories that both imitate expert behavior and respect all deployment constraints.

**Constraints.** To ensure reliable real-world execution, every generated trajectory must, at every time step, satisfy: **(1) Safety:** the state remains within a safe set $\mathbb{S}$ (e.g., avoid collisions; (**SC**)); **(2) Admissibility:** the action is within an admissible set $\mathbb{A}$ (e.g., satisfy torque or speed limits; (**AC**)); and **(3) Dynamic Consistency:** the trajectory obeys the system dynamics (**DC**). Neglecting any of these leads to unsafe, infeasible, or unrealizable plans: for example, trajectories may pass through obstacles (violating safety), demand unattainable actions (violating admissibility), or include state transitions that cannot be executed by the system (violating dynamics).

Formally, these requirements are posed on the distribution $p^{\theta}(\tau)$ of the learned planner as follows:

$$\forall \tau \sim p^{\theta}(\tau): \quad s(k) \in \mathbb{S}, \quad \forall k = 0, \ldots, H-1, \quad \textbf{(SC)}$$

$$\forall \tau \sim p^{\theta}(\tau): \quad a(k) \in \mathbb{A}, \quad \forall k = 0, \ldots, H-1, \quad \textbf{(AC)}$$

$$\forall \tau \sim p^{\theta}(\tau): \quad s(k) = f(s(k-1), a(k-1)),$$
$$\forall k = 1, \ldots, H-1. \qquad \textbf{(DC)}$$

**Objective.** Given expert data $\mathcal{D}$ and constraint sets $\mathbb{S}, \mathbb{A}$, our goal is to learn a generative model $p^{\theta}(\tau)$ that (i) matches the expert trajectory distribution, and (ii) ensures all sampled trajectories satisfy eqs. (**SC**), (**AC**), and (**DC**). This setting motivates methods that can flexibly enforce constraints, even as requirements change at test time.

## 4. Background: Learning to Plan with Flow Matching

When a trajectory data set $\mathcal{D}$ of an expert planner is given, Flow Matching (FM; Lipman et al. 2023; Zheng et al. 2023) is an effective technique to learn the distribution $p(\tau)$ of the data $\mathcal{D}$. In FM, the unknown distribution $p(\tau)$ is considered as the desired endpoint of a probability path $p_t^{\theta}(\tau)$, $t \in [0, 1]$. The remainder of the probability path $p_t^{\theta}(\tau)$ is characterized by a time-dependent vector field $v_t^{\theta} : [0, 1] \times \mathbb{R}^{(n+m)H} \to \mathbb{R}^{(n+m)H}$ parameterized by $\theta$ that acts on samples $\tau_0 \sim p_0(\tau)$ via the flow

$$\dot{\tau}_t = \frac{d}{dt}\tau_t = v_t^{\theta}(\tau_t), \qquad (2)$$

whereby the prior $p_0(\tau)$ is typically set to a Gaussian (Lipman et al., 2023). By prescribing a path from samples $\tau_0$ of $p_0(\tau)$ to data trajectories $\tau_1$ via a scheduled interpolation $\tau_t = \alpha(t)\tau_1 + \beta(t)\tau_0$ with $\alpha, \beta$ such that $\alpha(0) = 0$, $\beta(1) = 0$ and $\alpha(t) + \beta(t) = 1$, the distribution learning problem is transformed into the supervised learning problem

$$\mathcal{L}_{\text{CFM}}(\theta) = \qquad (3)$$
$$\mathbb{E}_{t \sim \mathcal{U}[0,1], \tau_t \sim p_t^{\theta}(\tau), \tau_1 \sim p(\tau)} ||v_t^{\theta}(\tau_t) - v_t(\tau_0, \tau_1)||_2^2,$$

where $v_t(\tau_0, \tau_1) = \dot{\alpha}(t)\tau_1 + \dot{\beta}(t)\tau_0$ follows from the interpolation. Minimizing this cost function via stochastic gradient descent, $v_t^{\theta}(\tau_t)$ can be efficiently trained using the trajectory data set $\mathcal{D}$.

Given an initial state $s_0$ and a trained vector field $v_t^{\theta}(\tau_t)$, we sample a random trajectory $\tau_0$ from the prior distribution $p_0$ and numerically solve the ordinary differential equation (ODE) in (2) using $\tau_0$ as the initial condition to obtain $\tau_1$. Thereby, the trajectories effectively become samples $\tau_1 \sim p^{\theta}(\tau)$, where $p^{\theta}(\tau)$ is implicitly represented through $v_t^{\theta}(\tau_t)$. While such samples can be directly used in planning, they generally do not satisfy the safety, admissibility, and dynamic consistency constraints in eqs. (**SC**), (**AC**), and (**DC**).

*Remark* 4.1. To allow plans with given initial states $s_0$, we only need to condition the initial distribution on $s(0) = s_0$ and ensure $\dot{\tau}_t^{s(0)} = 0$ for all $t \in [0, 1]$. For training, $s_0$ is chosen as the first state of trajectories in the data set, while arbitrary values can be set when sampling trajectories.

## 5. Control-Augmented Flow Matching for Constrained Planning

Despite the power of FM-based planners for trajectory generation, ensuring safety, admissibility, and dynamic

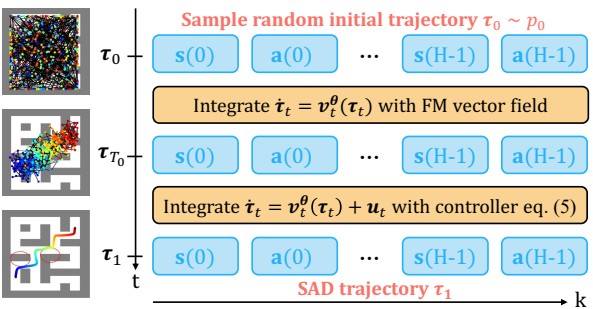

*Figure 1.* Overview of the trajectory generation with our pipeline.

consistency remains challenging, particularly under new test-time constraints. We address this with SAD-Flower, a control-augmented flow matching framework that provides formal guarantees for constraint satisfaction. In Section 5.1, we outline the control-theoretic intuition and its integration with FM. Section 5.2 details how constraint-aware quadratic programming augments sampling, and Section 5.3 presents theoretical guarantees of constraint satisfaction.

### 5.1. Control Augmentation for Safety, Admissibility, and Dynamic Consistency

To ensure the satisfaction of safety (**SC**), admissibility (**AC**) and dynamic consistency (**DC**), we extend the formulation in (2) at test time to a controlled dynamical system

$$\dot{\boldsymbol{\tau}}_t = \boldsymbol{v}_t^{\boldsymbol{\theta}}(\boldsymbol{\tau}_t) + \boldsymbol{u}_t, \qquad (4)$$

where the vector field $\boldsymbol{v}_t^{\boldsymbol{\theta}}(\boldsymbol{\tau}_t)$ represents the drift, while $\boldsymbol{u}_t$ is a control input. By choosing $\boldsymbol{u}_t = \boldsymbol{0}$, we recover standard flow matching as a special case of this formulation. Framing the problem in this way enables us to view the requirements in eqs. (**SC**), (**AC**), and (**DC**) as system properties, so that their satisfaction becomes a matter of control design with the following specifications.

**From State/Action Constraints to Barrier Specifications.** State and action constraints are set inclusion conditions, which require the controlled flow in (4) to converge to and subsequently maintain constraint satisfaction. This behavior can be formalized via control barrier functions (CBFs; Ames et al. 2017), whose level sets can encode the constraint sets $\mathbb{A}$ and $\mathbb{S}$. Thereby, we express state and action constraints as a condition on the growth of CBFs along the flow.

**From Dynamic Consistency to Lyapunov Specifications.** Dynamic consistency is an equality condition, whose violation needs to decay to 0 along the flow in (4). This property can be formalized using control Lyapunov functions (CLFs; Sontag 1983) – energy-like, non-negative functions with a minimum of 0 at the desired equilibrium. Hence, we formulate dynamic consistency as a condition on the decay of a CLF along the generation process of the flow.

**Prescribed-Time Specifications.** While FM simulates ODEs for time intervals $t \in [0,1]$, Lyapunov and barrier specifications usually relate to asymptotic guarantees with $t \to \infty$. This discrepancy necessitates the scheduling of a sufficiently fast growth of CBFs and decrease of CLFs along the flow, which corresponds to a prescribed-time specification for control (Song et al., 2017).

Our approach – SAD-Flower – splits the numerical integration of (4) into two phases as illustrated in Fig. 1. In the uncontrolled phase ($0 \le t < T_0$), trajectories evolve under the learned FM vector field without intervention ($\boldsymbol{u}_t = \boldsymbol{0}$) to preserve sample diversity (Fan et al., 2025). In the controlled phase ($T_0 \le t \le 1$), the control law $\boldsymbol{u}_t$ satisfying CLF, CBF, and prescribed-time specifications is applied when integrating (4). Thereby, the activation time $T_0 \in (0, 1)$ allows SAD-Flower to effectively balance generative flexibility with formal guarantees on safety (**SC**), admissibility (**AC**), and dynamic consistency (**DC**).

### 5.2. Control Design Using Control Lyapunov and Barrier Functions

Given the control specifications in Section 5.1, the actual control design problem remains. We first derive dedicated CBF and CLF constraints, which employ scheduling functions to ensure a sufficient growth/decrease rate. These constraints are exploited in an optimization-based control law.

**Control Barrier Constraints.** Since state/action constraints are specified separately for each time step, we design state CBFs $h_k^s(\boldsymbol{\tau}_t)$ and action CBFs $h_k^a(\boldsymbol{\tau}_t)$ for each time step $k = 0, \dots, H-1$. Each CBF itself is defined as a signed distance function (SDF; Park et al. 2019; Long et al. 2021), which measures the distance to the boundary of the set $\mathbb{S}$ and $\mathbb{A}$, respectively, and assigns a sign based on the inclusion in the constraint set.[1] Thus, these functions are only positive if states $\boldsymbol{s}(k)$ and actions $\boldsymbol{a}(k)$ are inside the sets $\mathbb{S}$ and $\mathbb{A}$, respectively. Non-negativity of the CBFs can be ensured by constraining the evolution of the CBF values along the flow, which results in the derivative condition

$$\dot{h}_k^s(\boldsymbol{\tau}_t) \ge -\varphi(t) h_k^s(\boldsymbol{\tau}_t), \quad \forall k = 1, \dots, H-1, \quad \text{(CBF-s)}$$

$$\dot{h}_k^a(\boldsymbol{\tau}_t) \ge -\varphi(t) h_k^a(\boldsymbol{\tau}_t), \quad \forall k = 0, \dots, H-1, \quad \text{(CBF-a)}$$

where $\dot{h}_k^{s/a}(\boldsymbol{\tau}_t) = \nabla^T h_k^{s/a}(\boldsymbol{\tau}_t) \left( \boldsymbol{v}_t^{\boldsymbol{\theta}}(\boldsymbol{\tau}_t) + \boldsymbol{u}_t \right)$ and $\varphi(t)$ is a scheduling function that we will design later.

**Control Lyapunov Constraints.** We define the CLF as the sum of squared consistency errors of a trajectory, i.e., $V(\boldsymbol{\tau}_t) = \frac{1}{2} \sum_{k=1}^{H-1} ||\boldsymbol{s}(k) - \boldsymbol{f}(\boldsymbol{s}(k-1), \boldsymbol{a}(k-1))||^2$.

This function is only 0 if the trajectory $\boldsymbol{\tau}_t$ is dynamically consistent, such that we constrain the evolution of its value

---

[1]Formal definition of $h_k^s(\boldsymbol{\tau}_t)$ and $h_k^a(\boldsymbol{\tau}_t)$ are in Appendix A

---

**Algorithm 1** Planning by SAD-Flower

---

1: **Input:** pretrained flow model $v_t^\theta(\tau_t)$
2: Initialize $\tau_0 \sim p_0(\tau)$
3: Solve ODE $\dot\tau_t = v_t^\theta(\tau_t)$ in $[0, T_0)$
4: Solve ODE $\dot\tau_t = v_t^\theta(\tau_t) + u_t$ in $[T_0, 1]$ with our controller (5) to get $\tau_1$
5: **Output:** SAD trajectory $\tau_1$

---

along the flow to be negative via

$$\dot{V}(\tau_t) \leq -\varphi(t)V(\tau_t), \qquad \text{(CLF)}$$

where $\dot{V}(\tau_t) = \nabla^T V(\tau_t)\left(v_t^\theta(\tau_t) + u_t\right)$ (Sontag, 1983) and $\varphi(t)$ is a scheduling function that we will design later. Computing $\nabla^T V(\tau_t)$ requires access to $f$. In some robotic domains, this information is available through high-fidelity simulators (Gaz et al., 2019; Howell et al., 2022; Acosta et al., 2022). If there exists no accurate physics-based simulator, e.g., for contact-rich manipulation tasks, a dynamics model can also be learned from the trajectory data. The learning errors directly correlate to the magnitude of dynamic consistency violations (Curi et al., 2020), such that practical consistency can still be achieved given a sufficiently precise learned model, and the (SC), (AC) can still be guaranteed, which is discussed in Appendix B.4.

**Prescribed-Time Scheduling.** To guarantee that the CBFs are positive and the CLF is 0 at the terminal time $t = 1$ regardless of its state at $t = T_0$, we employ a scheduling function $\varphi(t) = \frac{c}{(1-t)^2}$ with some constant $c > 0$. Due to the steep growth of $\varphi$ for $t \to 1$, the constraints in eqs. (CBF-s), (CBF-a) and (CLF) become increasingly more restrictive. This ensures that positivity of CBFs and a vanishing CLF are ensured at some time $t < 1$ (Song et al., 2023).

**Constrained Minimum-Norm Optimal Control.** To ensure the satisfaction of CLF and CBF constraints, we formulate the minimum-norm optimal control problem

$$u_t = \min_u ||u||^2 \quad \text{s.t. (CBF-s), (CBF-a) and (CLF)} \quad (5)$$

hold, which ensures them by construction, while simultaneously minimizing the perturbation of the learned vector field $v_t^\theta(\tau_t)$. Even though this optimization problem often consists of a large number of optimization variables and constraints, it can be solved comparatively efficiently since it is a quadratic program (QP). This renders the numerical integration of (4) with the control law in (5) computationally tractable when using dedicated QP solvers.

### 5.3. Theoretical Guarantees for Constraint Satisfaction and Consistency.

Due to the strong theoretical foundations of CBFs and CLFs, strong guarantees for safety, admissibility, and dynamic consistency can be provided in the following result.[2]

**Theorem 5.1.** *Assume that the QP in* (5) *is feasible for all* $t \in [T_0, 1]$. *Then, the solution* $\tau_t$ *of* (4) *with control law* $u_t = 0$ *for* $t < T_0$ *and* $u_t$ *defined in* (5) *for* $t \geq T_0$ *satisfies eqs.* (SC), (AC), *and* (DC) *at* $t = 1$ *for all initial conditions* $\tau_0$.

This result ensures that, under QP feasibility, SAD-Flower generates trajectories that are guaranteed to be safe, admissible, and dynamically consistent at the final time step.

*Remark* 5.2. While feasibility is a known challenge in general CBF-CLF based methods, it is typically not an issue in our framework. Due to the integrator-like structure of flow-matching dynamics, the individual CBF and CLF constraints are feasible under mild conditions (cf. Theorems B.1 and B.2). Infeasibility may occur only on measure-zero trajectory sets, which do not arise in practice during numerical integration of (4). Although slack-variable relaxations (Boyd & Vandenberghe, 2004) can be used as a fallback in other methods, they are not needed here, as our original QP remains feasible throughout all experiments.

*Remark* 5.3. When access to accurate system dynamics is limited, guarantees (SC) and (AC) can still be retained by incorporating robustness mechanisms such as robust CBFs or constraint tightening, as outlined in Appendix B.4.

*Remark* 5.4. Our control-theoretic formulation can be naturally applied to ODE-based generative samplers by treating the generative dynamics as a controlled system, as in (4). We focus on flow matching because its deterministic ODE form directly matches standard CBF/CLF tools. Since diffusion models also admit ODE-based samplers (Song et al., 2021), they can be handled by the same control layer. Thus, SAD-Flower can be viewed as a general constraint-enforcing control layer for ODE-based generative sampling.

## 6. Experiment

### 6.1. Experiment Setting

We evaluate all methods in 4 different benchmarks: Maze2d, Hopper, and Walker2d from the D4RL problem set (Fu et al., 2020), and Kuka Block-Stacking (Janner et al., 2022). [3]

- **Maze2d** (Fu et al., 2020) is a navigation task where a point mass is moved from an initial state to a specified goal. Actions are artificially constrained to $[-0.1, 0.1]$. State constraints are defined for two novel obstacles, which do not block feasible paths and therefore still allow the task to be completed. Training data is generated by a navigation planner for the given maze. We evaluate on two maze configurations: Large and Umaze.

---

[2]A proof and extended discussion of the theorem's assumptions can be found in Appendix B.

[3]Experimental details are provided in Appendix E.

*Table 2.* Performance of the proposed SAD-Flower and baselines across navigation, locomotion, and manipulation tasks. The methods are compared on the maximum safety (**SC**) and admissibility (**AC**) violations of planned trajectories, the dynamic consistency (**DC**) violation, and the model accuracy expressed through the reward. Truncate is not applicable in Maze2d (Umaze) due to more complex safety constraints, such that truncation becomes non-trivial (Xiao et al., 2025). Locomotion tasks are evaluated with two datasets: from a partially trained soft actor-critic policy (Haarnoja et al., 2018) (Medium, See Appendix C) and a mixture with expert data (Med-Expert). Lower constraint values (0 is perfect) indicate better adherence; higher reward indicates better performance. Bold denotes best results.

| Experiment | Metric | Diffuser | Trunc | CG | FM | S-Diffuser | D-Diffuser | Ours |
|---|---|---|---|---|---|---|---|---|
| Maze2d (Large) | safety | 0.43±0.39 | 0.10±0.25 | 0.27±0.37 | 0.37±0.39 | **0.00±0.00** | 0.83±0.24 | **0.00±0.00** |
| | admissib. | 0.09±0.01 | 0.11±0.05 | 0.12±0.03 | 0.13±0.01 | 0.09±0.02 | 0.13±0.04 | **0.00±0.00** |
| | dyn. consist. | 0.06±0.03 | 0.06±0.03 | 0.44±0.16 | 0.02±0.00 | 0.09±0.06 | 2.78±0.06 | **0.01±0.01** |
| | reward | 1.40±0.26 | 1.39±0.26 | 0.4±0.33 | 1.43±0.20 | 1.20±0.06 | 0.38±0.18 | **1.69±0.26** |
| Maze2d (Umaze) | safety | 0.04±0.20 | — | 0.51±0.33 | 0.11±0.22 | 0.87±3.77 | 0.05±0.15 | **0.00±0.00** |
| | admissib. | 0.11±0.02 | — | 0.10±0.01 | 0.09±0.01 | 0.14±0.02 | 0.09±0.02 | **0.00±0.00** |
| | dyn. consist. | 0.05±0.02 | — | 0.68±0.18 | **0.01±0.01** | 0.10±0.10 | 1.80±0.11 | **0.01±0.01** |
| | reward | 1.11±0.44 | — | 0.06±0.32 | 2.62±1.09 | 1.06±0.35 | 0.60±0.33 | **2.70±0.67** |
| Hopper (Med-Expert) | safety | 0.01±0.02 | 0.05±0.04 | 0.07±0.03 | 0.11±0.08 | 0.05±0.04 | 0.10±0.02 | **0.00±0.00** |
| | admissib. | 0.21±0.05 | 0.18±0.04 | 0.26±0.07 | 0.17±0.05 | 0.18±0.04 | **0.00±0.00** | **0.00±0.00** |
| | dyn. consist. | 0.38±0.03 | 0.41±0.04 | 0.79±0.10 | 0.23±0.02 | 0.36±0.06 | 0.16±0.01 | **0.01±0.01** |
| | reward | 1.06±0.18 | 0.50±0.12 | 0.73±0.022 | 1.02±0.20 | 0.53±0.19 | **1.12±0.01** | 0.93±0.23 |
| Walker2d (Med-Expert) | safety | 0.06±0.04 | 0.06±0.05 | 0.02±0.03 | 0.40±0.11 | 0.09±0.07 | 0.04±0.04 | **0.00±0.00** |
| | admissib. | 0.67±0.18 | 0.62±0.14 | 0.72±0.18 | 0.15±0.02 | 0.58±0.20 | **0.00±0.00** | **0.00±0.00** |
| | dyn. consist. | 0.71±0.05 | 0.72±0.05 | 0.83±0.91 | 0.44±0.01 | 0.79±0.03 | 0.69±0.05 | **0.04±0.04** |
| | reward | 1.06±0.23 | 0.56±0.29 | 0.39±0.19 | **1.07±0.01** | 0.59±0.21 | 0.95±0.24 | 0.89±0.32 |
| KUKA Block Stacking | safety | 0.23±0.09 | **0.00±0.00** | 0.22±0.09 | 0.02±0.04 | **0.00±0.00** | 0.14±0.13 | **0.00±0.00** |
| | reward | 0.46±0.23 | 0.45±0.21 | 0.45±0.23 | 0.44±0.20 | 0.49±0.23 | **0.55±0.26** | 0.45±0.21 |

- **Hopper and Walker2d** (Fu et al., 2020) are locomotion tasks where a one-legged and a bipedal robot must move forward by jumping and walking, respectively. Actions are constrained to the range $[-1, 1]$. State constraints are imposed by requiring the robot's torso center to remain below a prescribed height (default: 1.6), creating a conflict with the objective of fast forward movement.

- **Kuka Block-Stacking** (Janner et al., 2022) is a manipulation task where a 7-DOF robotic arm must stack a set of blocks. Unlike the other tasks, no action constraints or dynamic consistency requirements are enforced; only state constraints are applied, allowing us to study a variant of trajectory planning focused solely on state feasibility. These constraints ensure that self-collisions of the robot are avoided. Training data is generated using the PDDLStream planner (Garrett et al., 2020).

**Evaluation Metrics.** We evaluate the safety and admissibility violation of a trajectory via its maximal distance to the constraint sets $\mathbb{S}$ and $\mathbb{A}$, which we can effectively express through the CBFs as $-\min_k \min\{h_k^{s,a}(\boldsymbol{\tau}), 0\}$. Thus, a value of 0 means constraint satisfaction, while positive values imply a violation. Dynamical consistency is measured using the Lyapunov function, such that large values indicate inconsistency. The performance of planned trajectories is measured by D4RL normalized rewards and binary success rewards for stacking, as defined in (Janner et al., 2022).

**Baselines:** We compare our proposed SAD-Flower against the following generative planners:

- **Diffuser** (Janner et al., 2022) generates trajectories using a diffusion model, without considering safety, admissibility, or dynamic consistency.

- **Truncation (Trunc)** (Brockman et al., 2016) truncates diffusion-generated trajectories to satisfy constraints.

- **Classifier Guidance (CG)** (Dhariwal & Nichol, 2021) augments diffusion-based trajectory generation with constraint-gradient guidance during sampling.

- **Flow Matching (FM)** (Feng et al., 2025) trains a FM model as a baseline to generate trajectories without ensuring safety, admissibility, or dynamic consistency.

- **SafeDiffuser (S-Diffuser)** (Xiao et al., 2025) generates trajectories via diffusion, while projecting states onto the constraint sets using a CBF at each sampling step.

- **Decision Diffuser (D-Diffuser)** (Ajay et al., 2023) trains a diffusion model to generate state trajectories conditioned on constraints, with the corresponding actions inferred from a learned inverse dynamics model.

## 6.2. Constraint Satisfaction Across Tasks

As shown in Table 2, our method consistently satisfies safety and admissibility constraints while matching the planning performance of other methods across tasks. Although perfect dynamical consistency is not reached, the remaining violations are minor and mainly due to numerical integration of (4). These results demonstrate the effectiveness of our control-theoretic mechanism, which guarantees constraint satisfaction at test time. SAD-Flower achieves this performance at the cost of merely a small computation time increase compared to existing methods (Appendix, Table 20).

In the navigation tasks of Maze2D, SAD-Flower achieves high rewards while maintaining complete constraint satisfaction. Since the task goals and imposed constraints are not in conflict, our method effectively balances planning performance and constraint adherence. Among baselines, SafeDiffuser employs a constraint-following mechanism during the diffusion sampling process to enforce state constraints, but it fails to guarantee admissibility. In contrast, Diffuser and FM achieve high rewards at the cost of violations in both safety and admissibility.

In locomotion tasks, where dynamics are more complex, SAD-Flower is the only method that satisfies all constraints while maintaining strong dynamical consistency. Its slightly lower rewards stem from conflicts between the height constraint and the jumping/walking behaviors needed for high returns, making safety violations directly impact performance. For instance, CG achieves similar performance in Hopper (Med-Expert) when safety is respected. While some methods, such as Decision Diffuser, consistently ensure admissibility, all struggle with dynamical consistency due to the complexity of the robot dynamics.

In the Kuka Block-Stacking task, which excludes admissibility and dynamical consistency requirements, SAD-Flower leverages the simplified setting to guarantee safety while achieving rewards competitive with other safety-enforcing baselines. This highlights both the effectiveness and flexibility of our proposed approach.

We also compare with two constraint-aware baselines: CoBL (Mizuta & Leung, 2024) and reject-sampling (Appendix D). Reject-sampling filters candidate trajectories post-hoc, but fails under novel test-time constraints because the underlying generative model is trained without knowl-

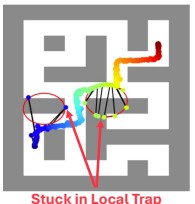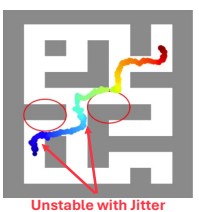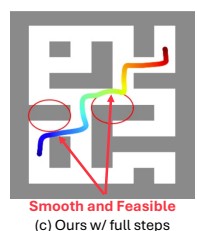


Stuck in Local Trap    Unstable with Jitter    Smooth and Feasible
(a) Safediffuser    (b) Ours w/ few ODE steps    (c) Ours w/ full steps


*Figure 2.* (a) Without enforcing dynamic consistency, applying state and action constraints can result in significant outliers, known as the local trap problem (Xiao et al., 2025). (b) Our method satisfies constraints, but using too few numerical integration steps for the ODE introduces jitter in the trajectory. (c) With sufficient integration steps, our method produces dynamically consistent trajectories while satisfying unseen constraints (red ellipses).

edge of these constraints, making valid samples unlikely to appear, as reported in Section 6.3. CoBL uses CBF/CLF-inspired reward shaping during denoising, but provides only soft guidance: it predicts actions only, does not explicitly handle action constraints, and uses CLFs for stability rather than dynamic consistency. In contrast, SAD-Flower jointly generates state-action trajectories and enforces all test-time constraints directly over the full trajectory.

## 6.3. Why Does SAD-Flower Work Effectively?

We analyze three key properties of SAD-Flower that contribute to its effectiveness.

**Dynamic Consistency Prevents Local Traps.** Projection-based constraint enforcement can introduce discontinuities in trajectories, causing the local trap problem (Xiao et al., 2025), as seen in Fig. 2(a) for SafeDiffuser in Maze2D. This arises from treating constraints independently at each step. In contrast, SAD-Flower enforces dynamic consistency by coupling consecutive states and actions through the CLF, ensuring coherent evolution during integration of the flow. This coupling prevents misaligned guidance and eliminates the risk of local traps, as demonstrated in Fig. 2(c). To verify the importance of this CLF term, we also ablate SAD-Flower

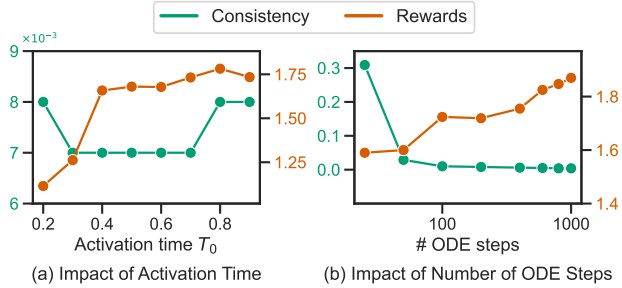


(a) Impact of Activation Time     (b) Impact of Number of ODE Steps


*Figure 3.* Effect of increasing the number of ODE steps for solving (4) and the activation time $T_0$ of controller (5) on the consistency and performance of trajectories.

*Table 3.* Benefits of QPs in SAD-Flower compared to general non-convex optimization of DPCC on Hopper (Med-Expert). The completed comparison is reported in Appendix Table 12.

| Methods | safety | admissib. | dyn. consist. | reward | comp. time [s] |
|---|---|---|---|---|---|
| DPCC | 0.01±0.03 | **0.00±0.00** | **0.01±0.01** | 0.61±0.07 | 4.24±1.64 |
| Ours | **0.00±0.00** | **0.00±0.00** | **0.01±0.01** | **0.93±0.23** | **0.06±0.00** |

*Table 4.* Performance of sampling-then-verification (reject-sample) method across all tasks. The completed comparison is reported at Table 13 in Appendix Section D.

| Experiment | safety | admissib. | dyn. consist. | reward |
|---|---|---|---|---|
| **Maze2d(Large)** | 0.16±0.29 | 0.11±0.00 | 0.02±0.01 | 1.52±0.28 |
| **Maze2d(UMaze)** | 0.04±0.09 | 0.10±0.01 | 0.01±0.01 | 2.91±0.65 |
| **Hopper(Med-Expert)** | 0.14±0.18 | 0.05±0.02 | 0.02±0.01 | 0.49±0.33 |
| **Walker2d(Med-Expert)** | 0.08±0.10 | 0.08±0.08 | 0.04±0.06 | 0.97±0.22 |
| **KUKA Block Stacking** | 0.01±0.01 | — | — | 0.44±0.67 |

by enforcing only safety and admissibility through CBFs. Without the CLF, planned actions may fail to realize the planned state trajectory, leading to dynamic inconsistency, safety violations, and local trapping behavior Fig. 2(a). This confirms that CLF-based dynamic-consistency enforcement is essential for reliable constrained planning.

**Delayed Activation Avoids Premature Interventions.** In the initial stages of flow sampling, trajectory distributions are relatively unstructured, and premature control can push trajectories away from learned behaviors. Perturbations introduced in this phase are propagated throughout the flow in (4), potentially degrading performance. SAD-Flower mitigates this by activating guidance only after a prescribed flow time $T_0$. As shown in Fig. 3 (left) for Maze2D, even a small $T_0$ achieves high rewards, while larger values can further improve performance with only minor increases in dynamic consistency violations. Safety and admissibility are satisfied across all $T_0$ values in our experiments (not shown due to space). These results highlight the benefit of our control-theoretic formulation, enabling delayed CLF and CBF activation without compromising constraint enforcement.

**QP vs. Non-Convex Optimization.** To assess the benefits of our QP-based formulation, we compare SAD-Flower with DPCC (Römer et al., 2025), which integrates optimal control into generative planning but applies full-system non-convex optimization from the earliest sampling stages. As shown in Table 3, both methods achieve comparable admissibility and dynamic consistency, but DPCC incurs much higher computation per sampling step and lower rewards, likely due to early optimization intervening before the generated samples develop meaningful structure. This highlights the efficiency of SAD-Flower, which achieves strong constraint satisfaction using QP solvers and delayed activation. Note though that the enforcement of constraints does still not come for free with QPs since the integration steps of the uncontrolled ODE merely take 0.01s on average. This overhead could be further reduced by recent advances in CBF-based optimization, such as closed-form solutions (Ong & Cortés, 2019) for some CBF-QPs or combining multiple CBF constraints into a single non-smooth CBF (Glotfelter et al., 2017), which may simplify the optimization and reduce runtime. While these techniques are not yet plug-and-play in general settings, they represent an active direction in control research and could be directly incorporated into SAD-Flower.

**Limitations of Naive Sampling-Then-Verification.** We further compare SAD-Flower with rejection-sampling, a naive sampling-then-verification strategy for constraint handling. This baseline first trains a flow matching model on the original dataset and, at inference time, samples a batch of trajectory candidates, discarding those that violate constraints. However, because the test-time constraints are unseen during training, the generative model has no mechanism to produce trajectories inside the new feasible set. As shown in Table 4, this leads to persistent safety or admissibility violations across tasks. In contrast, SAD-Flower works by explicitly enforcing the unseen constraints during inference, rather than hoping they appear in sampled candidates.

### 6.4. How Reliable Is SAD-Flower?

To highlight the reliability of SAD-Flower, we demonstrate its behavior when approaching the extremes in terms of approximations for implementation and problem difficulty.

**Reducing Integration Accuracy.** We first illustrate the reliability of SAD-Flower when the accuracy of integrating the controlled ODE in (4) is small, which we measure through the number of ODE discretization steps. As illustrated on the right of Fig. 3, increasing the number of ODE steps improves dynamic consistency since the controller in (5) has more opportunities to intervene. Even with as few as 25 ODE steps, the inconsistency is relatively small and corresponds only to minor jittering behavior in the

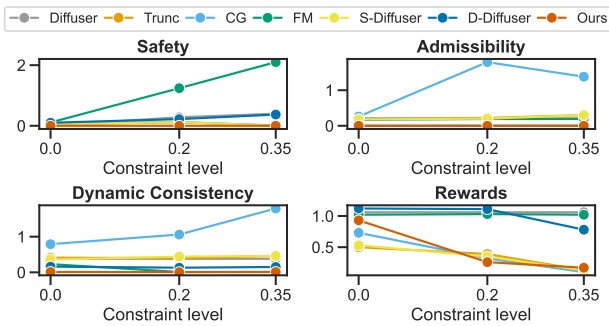

*Figure 4.* Performance of SAD-Flower and baselines under varying constraint tightening on Hopper (Med-Expert) benchmark.

*Table 5.* Performance of SAD-Flower and FM in a dexterous grasping scenario (Adroit-Hand for Relocate tasks).

| Methods | safety | admissib. | dyn. consist. | reward |
|---|---|---|---|---|
| **FM** | 0.15±0.21 | 0.62±0.19 | 0.07±0.04 | **1.07±0.08** |
| **Ours** | **0.00±0.00** | **0.00±0.00** | **0.06±0.09** | 1.05±0.23 |

*Table 7.* Effect of training dataset size on Hopper (Medium).

| Dataset | safety | admissib. | dyn. consist. | reward |
|---|---|---|---|---|
| 90% | 0.00 ± 0.00 | 0.00 ± 0.00 | 0.01 ± 0.02 | 0.35 ± 0.01 |
| 10% | 0.00 ± 0.00 | 0.00 ± 0.00 | 0.01 ± 0.01 | 0.38 ± 0.03 |
| 1% | 0.00 ± 0.00 | 0.00 ± 0.00 | 0.01 ± 0.01 | 0.34 ± 0.03 |
| 0.1% | 0.07 ± 0.03 | 0.00 ± 0.00 | 0.02 ± 0.01 | 0.37 ± 0.04 |
| 0.01% | 0.02 ± 0.09 | 0.00 ± 0.00 | 0.04 ± 0.02 | 0.36 ± 0.03 |

trajectories as illustrated in Fig. 2. With $\approx 100$ ODE steps, dynamic consistency is practically achieved. Safety and admissibility are ensured for the whole range of considered ODE steps (not depicted). In contrast, the rewards continue to grow for finer discretizations, which cause higher computational complexity. Thus, SAD-Flower allows to trade-off performance and complexity, while simultaneously ensuring constraint satisfaction and dynamic consistency.

**Handling Stricter Test-Time Constraints.** To evaluate the robustness of our method under unseen conditions, we test it with increasingly restrictive constraints by reducing the allowable torso height by 0.2 and 0.35 in the Hopper environment. As shown in Fig. 4, tighter constraints naturally reduce rewards due to their conflict with the task objective, a trend also observed in other state-constrained methods such as SafeDiffuser. Notably, the rewards of SAD-Flower are generally at least on par with these methods, even though the baselines exhibit significant admissibility and dynamic consistency violations. This clearly demonstrates the strong robustness of SAD-Flower in handling unseen test-time constraints.

**Scalability in High-Dimensional Tasks.** To assess the scalability, we evaluate SAD-Flower on the D4RL Adroit Relocate task, a dexterous manipulation benchmark (39D state and 30D action) that requires high-dimensional control. We impose state constraints via the arm's position for collision avoidance and action constraints through actuator limits. As shown in Table 5, standard flow matching achieves moderate success but violates constraints due to unconstrained sampling. In contrast, our approach consistently satisfies both state and action constraints, demonstrating its ability to scale to high-dimensional robotic systems.

**Sensitivity Analysis of Constraint-Enforcement Strength.** We conduct an ablation study on constraint strength parameter $c$ to assess the sensitivity of SAD-Flower. While the main experiments use $c = 0.5$, we vary it from 1.0 to 0.4.

*Table 6.* Sensitivity of the hyperparameter $c$ on Maze2d (Large).

| $c$ | safety | admissib. | dyn. consist. | reward |
|---|---|---|---|---|
| 1.0 | 0.00±0.00 | 0.00±0.00 | 0.02±0.01 | 1.65 ± 0.32 |
| 0.8 | 0.00±0.00 | 0.00±0.00 | 0.02±0.01 | 1.57 ± 0.25 |
| 0.6 | 0.00±0.00 | 0.00±0.00 | 0.02±0.01 | 1.52 ± 0.26 |
| 0.4 | 0.00±0.00 | 0.00±0.00 | 0.01 ± 0.01 | 1.50 ± 0.28 |

As shown in Table 6, SAD-Flower satisfies all constraints across all values. Violations of dynamic consistency remain negligible and stable, while task performance varies moderately. These results demonstrate that SAD-Flower achieves constraint satisfaction without requiring precise hyperparameter tuning, highlighting its practical reliability.

**Robustness Under Imperfect Dynamics Models.** To evaluate robustness under model approximation error, we trained the forward model on progressively smaller subsets of the Hopper-Medium dataset. As shown in Table 7, SAD-Flower maintains full constraint satisfaction even with only 1% of the original data, and dynamic consistency remains stable. Degradation becomes apparent only when the dataset is reduced to 0.1%, at which point safety violations emerge. These results show that SAD-Flower is tolerant to moderately inaccurate dynamics models, so training a sufficiently accurate model is practical, while constraint satisfaction can degrade when the learned dynamics model becomes highly inaccurate. We further confirm this trend through a Gaussian-noise ablation on Maze dynamics in Appendix C. These results characterize how model accuracy affects SAD-Flower: moderate errors can be tolerated, but severe model mismatch can degrade dynamic consistency and constraint satisfaction.

## 7. Conclusion

We presented SAD-Flower, a control-augmented flow matching framework that ensures safe, admissible, and dynamically consistent trajectory planning. By reformulating flow matching as a controllable dynamical system with a virtual control input, our method leverages Control Barrier Function and Control Lyapunov Function conditions scheduled using prescribed-time control principles to enforce constraints at test time without retraining. Experiments across multiple tasks show that SAD-Flower achieves perfect constraint satisfaction, avoids local traps, and maintains competitive task performance. These results establish it as a practical, theoretically grounded solution for real-world deployment. Looking ahead, extending SAD-Flower to stochastic dynamics, complex constraint geometries, and online replanning offers promising directions for safe and reliable generative trajectory planning.

## Impact Statement

This work introduces SAD-Flower, a control-augmented flow matching framework that enables generative models to plan trajectories with formal guarantees on safety, admissibility, and dynamic consistency. Our primary contribution is methodological, aiming to advance the field of safe and constraint-aware generative modeling in robotics and planning. The proposed framework has potential applications in domains where safety is critical, such as autonomous navigation, robotic manipulation, and assistive systems.

While our method improves constraint satisfaction and robustness at test time, it still inherits limitations common to data-driven approaches. In particular, discrepancies between the training data distribution and the deployment environment may lead the model to reproduce biased or suboptimal behaviors. In addition, although SAD-Flower is designed for planning rather than real-time control, further optimization of inference speed may be required before deployment in safety-critical systems with strict real-time requirements. We therefore recommend that applications in such domains be accompanied by rigorous validation, system-level testing, and appropriate human oversight.

We see this work as a step toward safer generative planning and believe it can contribute positively to automation in domains that currently rely on hand-crafted rule sets for constraint enforcement. However, as with many technologies in robotics and AI, broader deployment may impact labor markets and should be guided by thoughtful policy and ethical frameworks.

## Acknowledgement

We thank Ziyi Chen and Bryan Daniel Umbarila Rubiano for the simulation setup. We gratefully acknowledge the support of DAAD programme Konrad Zuse Schools of Excellence in Artificial Intelligence, sponsored by the Federal Ministry of Research, Technology and Space, European Union's Horizon Europe innovation action programme under grant agreement No. 101093822, "SeaClear2.0", research training group METEOR DFG (GRK 3081) funded by German Research Foundation (DFG), Federal Ministry of Research, Technology, and Space (BMFTR) as part of the research program Communication Systems "Souverän. Digital. Vernetzt.", joint project 6G-life under project identification number 16KIS2414, and a start-up grant of the National University of Singapore. This work was supported in part by the National Science and Technology Council, Taiwan, under Grants 114-2628-E-002-021- and 115-2634-F-002 -012-, and the Taiwan Centers of Excellence in Artificial Intelligence. Shao-Hua Sun was supported by the Yushan Fellow Program of the Ministry of Education, Taiwan.

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

# Appendix

## A. Signed Distance Functions

Given sets $\mathbb{S}$ and $\mathbb{A}$, the signed distance functions are defined as the minimal distance to a point on the boundary $\partial\mathbb{S}$ and $\partial\mathbb{A}$ of the sets, with the sign indicating if the state/action along the trajectory is in the corresponding sets. This formally leads to the following functions

$$h_k^s(\boldsymbol{\tau}) = \begin{cases} \min_{\boldsymbol{s}\in\partial\mathbb{S}}||\boldsymbol{s}(k)-\boldsymbol{s}|| & \text{if } \boldsymbol{s}\in\mathbb{S} \\ -\min_{\boldsymbol{s}\in\partial\mathbb{S}}||\boldsymbol{s}(k)-\boldsymbol{s}|| & \text{else} \end{cases} \qquad h_k^a(\boldsymbol{\tau}) = \begin{cases} \min_{\boldsymbol{a}\in\partial\mathbb{A}}||\boldsymbol{a}(k)-\boldsymbol{a}|| & \text{if } \boldsymbol{a}\in\mathbb{A} \\ -\min_{\boldsymbol{a}\in\partial\mathbb{A}}||\boldsymbol{a}(k)-\boldsymbol{a}|| & \text{else.} \end{cases} \tag{6}$$

## B. Proofs of Theoretical Results

### B.1. Feasibility of CBF Constraints

**Theorem B.1.** *Assume that $\varphi(t) = \frac{c}{(t-1)^2}$ with constant $c \in \mathbb{R}_+$. If $h_k^s$ and $h_k^a$ are differentiable, then, for each $\boldsymbol{\tau}$, there exists a $\boldsymbol{u}$ jointly satisfying* (CBF-s) *and* (CBF-a).

*Proof.* Since the constraints defined in eqs. (CBF-s) and (CBF-a) concern independent variables, individual feasibility of each constraint implies joint feasibility of all of them. Due to the assumed differentiability of $h_k^{s,a}$ and the definition of SDFs implying $||\nabla h_k^{s,a}(\boldsymbol{\tau})|| = 1$, $\boldsymbol{u}$ can always be chosen such $\nabla^T h_k^{s,a}(\boldsymbol{\tau})\boldsymbol{u}$ takes an arbitrary value. Thus, the constraints defined in eqs. (CBF-s) and (CBF-a) are always feasible. $\square$

The assumption on the differentiability of $h_k^s$ and $h_k^a$ is required as the signed distance functions (SDF) (6) are generally not smooth. However, they are differentiable almost everywhere for sets with smooth boundary under weak assumptions (Gilbarg et al., 1977). Non-differentiable set boundaries can be overcome via smoothing transformations (Begzadić et al., 2025) or by increasing the number of CBF constraints via a decomposition of $\bar{\mathbb{S}}$ and $\bar{\mathbb{A}}$ into suitable subsets $\bar{\mathbb{S}}_i$ and $\bar{\mathbb{A}}_i$, such that the SDFs can be computed with respect to the subsets instead. For example, this approach immediately yields linear functions of the form $h_{k,i}^a(\boldsymbol{\tau}) = \bar{a} \pm \boldsymbol{a}_i(k)$ for individual elements $\boldsymbol{a}_i(k)$ of actions when defining suitable subsets $\bar{\mathbb{A}}_i$ for the the commonly employed box constraints $||\boldsymbol{a}_i(k)||_\infty \le \bar{a}$ with some constant $a \in \mathbb{R}_+$. Thus the differentiability assumption in Theorem B.1 is rather technical and often not relevant in practice for the usage of SDFs in CBF constraints (Long et al., 2021).

### B.2. Feasibility of CLF Constraints

**Theorem B.2.** *Assume that $\varphi(t) = \frac{c}{(t-1)^2}$ with constant $c \in \mathbb{R}_+$. If $\{(\boldsymbol{s},\boldsymbol{a}) : \text{rank}(\frac{\partial}{\partial\boldsymbol{s}}\boldsymbol{f}(\boldsymbol{s},\boldsymbol{a})) \neq n \wedge \text{rank}(\frac{\partial}{\partial\boldsymbol{a}}\boldsymbol{f}(\boldsymbol{s},\boldsymbol{a})) \neq m\} = \emptyset$, then, for each $\boldsymbol{\tau}$, there exists a $\boldsymbol{u}$ satisfying* (CLF).

*Proof.* Let

$$\boldsymbol{\ell}_k(\tau) = \left(\boldsymbol{\tau}^{\boldsymbol{s}(k+1)} - \boldsymbol{f}(\boldsymbol{\tau}^{\boldsymbol{s}(k)}, \boldsymbol{\tau}^{\boldsymbol{a}(k)})\right). \tag{7}$$

Then, the gradient of $V$ is given by

$$\nabla V(\boldsymbol{\tau}) = \begin{bmatrix} -\frac{\partial\boldsymbol{f}(\boldsymbol{\tau}^{\boldsymbol{s}(0)},\boldsymbol{\tau}^{\boldsymbol{a}(0)})}{\partial\boldsymbol{\tau}^{\boldsymbol{s}(0)}}\boldsymbol{\ell}_0(\boldsymbol{\tau}) \\ -\frac{\partial\boldsymbol{f}(\boldsymbol{\tau}^{\boldsymbol{s}(0)},\boldsymbol{\tau}^{\boldsymbol{a}(0)})}{\partial\boldsymbol{\tau}^{\boldsymbol{a}(0)}}\boldsymbol{\ell}_0(\boldsymbol{\tau}) \\ \boldsymbol{\ell}_0(\boldsymbol{\tau}) - \frac{\partial\boldsymbol{f}(\boldsymbol{\tau}^{\boldsymbol{s}(1)},\boldsymbol{\tau}^{\boldsymbol{a}(1)})}{\partial\boldsymbol{\tau}^{\boldsymbol{s}(1)}}\boldsymbol{\ell}_1(\boldsymbol{\tau}) \\ -\frac{\partial\boldsymbol{f}(\boldsymbol{\tau}^{\boldsymbol{s}(1)},\boldsymbol{\tau}^{\boldsymbol{a}(1)})}{\partial\boldsymbol{\tau}^{\boldsymbol{a}(1)}}\boldsymbol{\ell}_1(\boldsymbol{\tau}) \\ \vdots \\ \boldsymbol{\ell}_{H-2}(\boldsymbol{\tau}) - \frac{\partial\boldsymbol{f}(\boldsymbol{\tau}^{\boldsymbol{s}(H-1)},\boldsymbol{\tau}^{\boldsymbol{a}(H-1)})}{\partial\boldsymbol{\tau}^{\boldsymbol{s}(H-1)}}\boldsymbol{\ell}_{H-1}(\boldsymbol{\tau}) \\ -\frac{\partial\boldsymbol{f}(\boldsymbol{\tau}^{\boldsymbol{s}(H-1)},\boldsymbol{\tau}^{\boldsymbol{a}(H-1)})}{\partial\boldsymbol{\tau}^{\boldsymbol{a}(H-1)}}\boldsymbol{\ell}_{H-1}(\boldsymbol{\tau}) \\ \boldsymbol{\ell}_{H-1}(\boldsymbol{\tau}) \\ \boldsymbol{0} \end{bmatrix}. \tag{8}$$

In the following, we will show by contradiction that $\nabla V(\boldsymbol{\tau}) = \mathbf{0}$ if and only if $\boldsymbol{\ell}_k(\boldsymbol{\tau}) = \mathbf{0}$ for all $k = 0, \dots, H-1$. For this purpose, assume that $\nabla V(\boldsymbol{\tau}) = \mathbf{0}$ and $\boldsymbol{\ell}_i \neq \mathbf{0}$ for some $i = 1, \dots, H-1$. If $\mathrm{rank}(\frac{\partial \boldsymbol{f}(\boldsymbol{\tau}^{\boldsymbol{s}(i)}, \boldsymbol{\tau}^{\boldsymbol{a}(i)})}{\partial \boldsymbol{\tau}^{\boldsymbol{a}(i)}}) \neq m$, $\nabla V(\boldsymbol{\tau}) \neq \mathbf{0}$ is trivially contradicted. If $\mathrm{rank}(\frac{\partial \boldsymbol{f}(\boldsymbol{\tau}^{\boldsymbol{s}(i)}, \boldsymbol{\tau}^{\boldsymbol{a}(i)})}{\partial \boldsymbol{\tau}^{\boldsymbol{s}(i)}}) \neq n$, $\nabla V(\boldsymbol{\tau}) = \mathbf{0}$ requires $\boldsymbol{\ell}_{i-1} \neq \mathbf{0}$. Hence, we can consider the same two cases as for $\boldsymbol{\ell}_i \neq \mathbf{0}$. By repeating this procedure and always considering the case $\mathrm{rank}(\frac{\partial \boldsymbol{f}(\boldsymbol{\tau}^{\boldsymbol{s}(k)}, \boldsymbol{\tau}^{\boldsymbol{a}(k)})}{\partial \boldsymbol{\tau}^{\boldsymbol{s}(k)}}) \neq n$, we eventually end up with the condition

$$\frac{\partial \boldsymbol{f}(\boldsymbol{\tau}^{\boldsymbol{s}(0)}, \boldsymbol{\tau}^{\boldsymbol{a}(0)})}{\partial \boldsymbol{\tau}^{\boldsymbol{s}(0)}} \boldsymbol{\ell}_0(\boldsymbol{\tau}) = \mathbf{0} \qquad \wedge \qquad \mathrm{rank}(\frac{\partial \boldsymbol{f}(\boldsymbol{\tau}^{\boldsymbol{s}(0)}, \boldsymbol{\tau}^{\boldsymbol{a}(0)})}{\partial \boldsymbol{\tau}^{\boldsymbol{a}(0)}}) \neq n \qquad \wedge \qquad \boldsymbol{\ell}_0 \neq \mathbf{0}, \qquad (9)$$

which cannot be satisfied. Thus, $\nabla V(\boldsymbol{\tau}) \neq \mathbf{0}$ holds if $\boldsymbol{\ell}_k(\boldsymbol{\tau}) \neq \mathbf{0}$ for some $k$. Consequently, there always exists a $\boldsymbol{u}$ such that (CLF) is satisfied, rendering the constraint feasible. $\qquad \square$

To ensure the feasibility of CLF constraints, we again need one technical assumption: Through infinitesimal changes of states or actions, the dynamics $\boldsymbol{f}$ can be changed in arbitrary directions. If this property is not satisfied, $\nabla V(\boldsymbol{\tau})$ does not necessarily provide information about directions for reducing the violation of (DC). Note that matrices with rank deficiency have zero measure among all matrices, such that infeasibilities related to a violation of the rank condition in Theorem B.2 occur only at isolated states (except for special cases of dynamics, e.g., piecewise constant dynamics). Thus, the rank condition is usually satisfied almost everywhere for many relevant systems, which is sufficient for the feasibility of the CLF constraint in (CLF) in practice.

## B.3. Proof of Theorem 5.1

*Proof of Theorem 5.1.* Due to the assumed feasibility of constraints, (4) controlled by (5) satisfies (CBF-s), (CBF-a), and (CLF), i.e., prescribed-time CBF and CLF conditions are satisfied. As $h_k^{s,a}$ is positive inside $\mathbb{S}/\mathbb{A}$, negative outside, and zero on the boundaries of these sets, it corresponds to a prescribed-time control barrier function (Huang et al., 2024b). Therefore, it immediately follows from (Huang et al., 2024b, Theorem 1) that $h_k^{s,a}(\boldsymbol{\tau}_1) > 0$ at $t = 1$, which implies satisfaction of safety (SC) and admissibility (AC) by construction. Since $V$ is positive definite, it corresponds to a prescribed-time control Lyapunov function (Song et al., 2019). Hence, it immediately follows from (Song et al., 2017, Theorem 2) that $V(\boldsymbol{\tau}_1) = 0$, which implies satisfaction of (DC) by construction. $\qquad \square$

While the two constraints in (CBF-s) and (CBF-a) concern independent variables, the addition of constraint (CLF) introduces a coupling between the constraints. This coupling can cause infeasibility at some trajectories $\boldsymbol{\tau}$ in general, but technical conditions exist that exclude them (Wang et al., 2024). Moreover, these infeasibilities are not an issue from a practical perspective since they often affect isolated $\boldsymbol{\tau}$ or small subsets of trajectories usually not occurring while numerically integrating (4).

## B.4. Learning Error of the Dynamics Model

When the true system dynamics $\boldsymbol{f}^*$ are unknown, SAD-Flower relies on a learned forward model $\boldsymbol{f}$ trained from data. This inevitably introduces approximation error, which affects the predicted next states during sampling. To quantify the deviation between the learned and true dynamics, we begin with the following result.

**Lemma B.3** (State Deviation Under Model Approximation). *If the forward model $\boldsymbol{f}$ has Lipschitz constant $L_f$, and the pointwise approximation error to the true dynamics $\boldsymbol{f}^*$ is bounded as*

$$||\boldsymbol{f}^*(\boldsymbol{s}(k), \boldsymbol{a}(k)) - \boldsymbol{f}(\boldsymbol{s}(k), \boldsymbol{a}(k))|| \leq \zeta \qquad (10)$$

*Then the deviation between the true trajectory $\boldsymbol{s}^*(k)$ and the learned trajectory $\boldsymbol{s}(k)$ after $k$ steps satisfies*

$$||\boldsymbol{s}^*(k) - \boldsymbol{s}(k)|| \leq \zeta \sum_{i=0}^{k-1} L_f^i =: \xi. \qquad (11)$$

*Proof.* Let the true and learned states evolve as:

$$\boldsymbol{s}^*(k) = \boldsymbol{f}^*(\boldsymbol{s}^*(k-1), \boldsymbol{a}(k-1)), \quad \boldsymbol{s}(k) = \boldsymbol{f}(\boldsymbol{s}(k-1), \boldsymbol{a}(k-1)).$$

Then,

$$\begin{aligned}
||\boldsymbol{s}^*(k) - \boldsymbol{s}(k)|| &= ||\boldsymbol{f}^*(\boldsymbol{s}^*(k-1), \boldsymbol{a}(k-1)) - \boldsymbol{f}(\boldsymbol{s}(k-1), \boldsymbol{a}(k-1))|| \\
&\leq ||\boldsymbol{f}^*(\boldsymbol{s}^*(k-1), \boldsymbol{a}(k-1)) - \boldsymbol{f}(\boldsymbol{s}^*(k-1), \boldsymbol{a}(k-1))|| \\
&+ ||\boldsymbol{f}(\boldsymbol{s}^*(k-1), \boldsymbol{a}(k-1)) - \boldsymbol{f}(\boldsymbol{s}(k-1), \boldsymbol{a}(k-1))|| \\
&\leq \zeta + L_f ||\boldsymbol{s}^*(k-1) - \boldsymbol{s}(k-1)|| \\
&\leq \xi,
\end{aligned} \tag{12}$$

Unrolling this recursion yields the desired bound. $\qquad\square$

This result gives an explicit upper bound $\xi$ on the deviation between the true and learned trajectories. The Lipschitz constant of the learned forward model $L_f$ and the error bound of forward dynamics $\zeta$ can be obtained with several techniques, e.g., automatic differentiation algorithms (Virmaux & Scaman, 2018), convex optimization (Fazlyab et al., 2019), Gaussian process (Lederer et al., 2019). In principle, this bound can be leveraged to ensure safety and admissibility under model approximation error by adopting a robust Control Barrier Function (RCBF) formulation (Buch et al., 2021), which is applied in our implementation. Specifically, the CBF condition (CBF-s) can be modified as

$$\dot{h}_k^s(\boldsymbol{\tau}_t) \geq -\varphi(t) h_k^s(\boldsymbol{\tau}_t) - (\varphi(t) L_h + ||\boldsymbol{u}_t|| L_{\nabla h}) \xi, \quad \forall k = 1, \dots, H-1, \tag{RCBF-s}$$

where $L_h$ and $L_{\nabla h}$ are Lipschitz constants of $h_k^s(\boldsymbol{\tau}_t)$ and its gradient, respectively.

Under this robust formulation, the corresponding control input for $t \geq T_0$ would be obtained by solving

$$\boldsymbol{u}_t = \min_{\boldsymbol{u}} ||\boldsymbol{u}||^2 \qquad \text{s.t. eqs. (RCBF-s), (CBF-a) and (CLF) hold,} \tag{13}$$

If such a controller were applied (with $\boldsymbol{u}_t = 0$ for $t < T_0$), the trajectory $\boldsymbol{\tau}_t^*$ generated by the *true* dynamics satisfies the original safety and admissibility constraints (SC) and (AC). Although dynamic consistency cannot be formally guaranteed without access to the true system $\boldsymbol{f}^*$, the robust CBF formulation provides a principled method for enforcing constraint satisfaction in the presence of model approximation error. Introducing norm-dependent terms in (RCBF-s) transforms the QP into a second-order cone program (SOCP), which may incur slightly higher computational cost. Nevertheless, the problem remains efficiently solvable in practice and offers strong guarantees in return.

An alternative strategy for addressing model approximation error is to apply conservative constraint tightening—shrinking the constraint sets by a safety margin to account for prediction uncertainty, as in Römer et al. (2025). While this approach is often simpler to implement than robust CBFs, our experiments showed that constraint satisfaction remained reliable without additional tightening. Nonetheless, both constraint tightening and robust CBFs remain viable options, particularly in scenarios with higher model uncertainty.

## C. Additional Experiment Details and Results

**Generalization Across Datasets.** We evaluate SAD-Flower on locomotion tasks using flow models trained on different datasets in the Hopper and Walker2d environments (e.g., the Medium dataset). As shown in Table 8, our method consistently ensures perfect satisfaction of both safety and admissibility constraints, regardless of the dataset used to train the underlying flow model. These results demonstrate the generalization ability of SAD-Flower when applied to different pre-trained generative planners.

**Robustness to Stricter Test-Time Constraints.** We evaluate the robustness of SAD-Flower by tightening the test-time constraint on torso height (Fig. 5) in the Hopper environment, decreasing the upper bound from 1.6 to 1.4 and 1.25 settings not encountered during training. As shown in Table 9, our method maintains perfect satisfaction of both safety and admissibility constraints across all levels of difficulty, despite the increasingly restrictive conditions. This demonstrates SAD-Flower's strong generalization to stricter, unseen constraints at test time. While task rewards decrease modestly—as expected due to the heightened challenge—our method consistently preserves formal guarantees of constraint satisfaction, validating the effectiveness of prescribed-time control as a flexible enforcement mechanism even under distributional shifts.

*Table 8.* Performance of the proposed SAD-Flower and baselines for locomotion tasks with medium-expert dataset. The methods are compared on the maximum safety and admissibility constraint violations of planned trajectories, the magnitude of dynamic consistency violation, and the model accuracy expressed through the reward.

| Experiment | Metric | Diffuser | Trunc | CG | FM | S-Diffuser | D-Diffuser | Ours |
|---|---|---|---|---|---|---|---|---|
| Hopper (Medium) | safety | 0.01±0.02 | 0.05±0.03 | **0.00±0.00** | 0.39±0.13 | 0.01±0.01 | 0.15±0.01 | **0.00±0.00** |
| | admissib. | 0.21±0.05 | 0.18±0.04 | 0.16±0.03 | 0.32±0.21 | 0.18±0.04 | **0.00±0.00** | **0.00±0.00** |
| | dyn. consist. | 0.46±0.01 | 0.47±0.01 | 0.95±0.02 | 0.42±0.39 | 0.47±0.01 | 0.18±0.01 | **0.01±0.01** |
| | reward | 0.44±0.05 | 0.45±0.06 | 0.39±0.03 | **0.49±0.05** | 0.45±0.06 | 0.48±0.08 | 0.34±0.03 |
| Walker2d (Medium) | safety | 0.03±0.03 | 0.02±0.01 | 0.02±0.02 | 0.21±0.15 | 0.02±0.02 | 0.09±0.04 | **0.00±0.00** |
| | admissib. | 0.56±0.10 | 0.44±0.19 | 0.54±0.14 | 0.48±0.06 | 0.52±0.12 | **0.00±0.00** | **0.00±0.00** |
| | dyn. consist. | 0.68±0.08 | 0.65±0.08 | 0.72±0.38 | 0.40±0.06 | 0.64±0.07 | 1.52±0.05 | **0.07±0.15** |
| | reward | 0.57±0.26 | 0.50±0.26 | 0.55±0.28 | 0.73±0.15 | 0.49±0.23 | **0.76±0.16** | 0.42±0.23 |

*Table 9.* Performance of the proposed SAD-Flower and baselines depending on constraint tightness. Constraints for the Hopper are tightened via lower admissible heights.

| Experiment | Metric | Diffuser | Truncate | CG | FM | S-Diffuser | D-Diffuser | Ours |
|---|---|---|---|---|---|---|---|---|
| Hopper (height=1.6) | safety | 0.01±0.02 | 0.05±0.04 | 0.07±0.03 | 0.11±0.08 | 0.05±0.04 | 0.10±0.02 | **0.00±0.00** |
| | admissib. | 0.21±0.05 | 0.18±0.04 | 0.26±0.07 | 0.17±0.05 | 0.18±0.04 | **0.00±0.00** | **0.00±0.00** |
| | dyn. consist. | 0.38±0.03 | 0.41±0.04 | 0.79±0.10 | 0.23±0.02 | 0.36±0.06 | 0.16±0.01 | **0.01±0.01** |
| | reward | 1.06±0.18 | 0.50±0.12 | 0.73±0.022 | 1.02±0.20 | 0.53±0.19 | **1.12±0.01** | 0.93±0.23 |
| Hopper (height=1.4) | safety | 0.28±0.08 | 0.12±0.09 | 0.04±0.03 | 1.24±0.07 | 0.12±0.09 | 0.22±0.03 | **0.00±0.00** |
| | admissib. | 0.21±0.05 | 0.22±0.09 | 1.79±0.33 | 0.19±0.10 | 0.20±0.08 | **0.00±0.00** | **0.00±0.00** |
| | dyn. consist. | 0.38±0.03 | 0.40±0.08 | 1.06±0.03 | 0.01±0.01 | 0.44±0.06 | 0.13±0.01 | **0.01±0.01** |
| | reward | 1.06±0.18 | 0.39±0.17 | 0.32±0.15 | 1.03±0.17 | 0.35±0.10 | **1.11±0.02** | 0.26±0.02 |
| Hopper (height=1.25) | safety | 0.40±0.04 | **0.00±0.00** | 0.01±0.01 | 2.10±0.07 | **0.00±0.00** | 0.37±0.02 | **0.00±0.00** |
| | admissib. | 0.21±0.05 | 0.30±0.05 | 1.38±0.27 | 0.05±0.10 | 0.30±0.05 | **0.00±0.00** | **0.00±0.00** |
| | dyn. consist. | 0.38±0.03 | 0.46±0.21 | 1.79±0.08 | **0.01±0.01** | 0.46±0.21 | 0.15±0.01 | **0.01±0.01** |
| | reward | 1.06±0.18 | 0.13±0.02 | 0.10±0.05 | **1.02±0.16** | 0.13±0.02 | 0.78±0.02 | 0.17±0.00 |

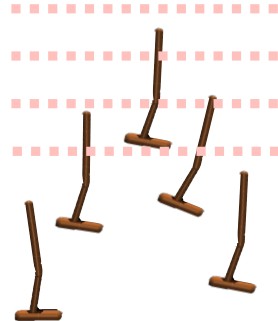

*Figure 5.* Performance under progressively tightened constraints in the locomotion task. As the allowed torso height decreases, the admissibility constraint becomes stricter, creating a conflict with the task objective of jumping forward.

**Comparison with DPCC (Römer et al., 2025).** To provide a more comprehensive comparison with optimization-based generative planning, we evaluate SAD-Flower against three DPCC variants (Römer et al., 2025): DPCC-T, which selects samples with minimal deviation from the previous timestep; DPCC-C, which minimizes cumulative projection cost; and DPCC-R, which selects samples randomly after projection. All DPCC variants leverage full-system nonlinear optimization at

*Table 10.* Performance of SAD-Flower and all baselines in a dexterous grasping scenario (Adroit-Hand for Relocate tasks).

| Methods | safety | admissib. | dyn. consist. | reward |
|---|---|---|---|---|
| **Diffuser** | 2.53±0.78 | 0.32±0.07 | 0.09±0.04 | 1.04±0.07 |
| **Trunc** | **0.00±0.00** | 0.32±0.07 | 0.11±0.03 | 1.04±0.06 |
| **CG** | 2.51±0.77 | 0.31±0.07 | 0.15±0.06 | 1.03±0.06 |
| **S-Diffuser** | **0.00±0.00** | 0.30±0.06 | 0.13±0.05 | 1.01±0.06 |
| **D-Diffuser** | 2.21±0.47 | 0.23±0.06 | 0.23±0.08 | 1.06±0.07 |
| **FM** | 0.15±0.21 | 0.62±0.19 | 0.07±0.04 | **1.07±0.08** |
| **Ours** | **0.00±0.00** | **0.00±0.00** | **0.06±0.09** | 1.05±0.23 |

*Table 11.* Performance when injecting noise into the dynamics.

| **No noise** | $\mathcal{N}(0, 0.1)$ | $\mathcal{N}(0, 0.2)$ | $\mathcal{N}(0, 0.4)$ |
|---|---|---|---|

each denoising step to enforce constraints. As shown in Table 12, while all DPCC variants succeed in satisfying admissibility and achieving dynamic consistency, they exhibit consistently higher safety violations (up to 0.02) even when we apply the constraint tightening technique (using height=1.5 under the constraint with height=1.6) and incur significantly greater computational cost—up to 70× slower than SAD-Flower. These issues likely stem from performing optimization over early-stage noisy samples, which may be difficult or even infeasible to correct without distorting future structure. Furthermore, the lower planning reward across all DPCC variants (0.61–0.65 vs. 0.93) suggests that early and aggressive control leads to sample deviation from the distribution learned by the generative model. In contrast, SAD-Flower activates control later using a lightweight QP-based formulation and preserves both safety and performance, highlighting the advantages of flexible, prescribed-time constraint enforcement.

**Performance of Adroit Task Across All Baselines** In addition to the comparison with standard flow matching in the main text, we evaluate SAD-Flower against all baselines on the D4RL Adroit Relocate task. This benchmark involves a 39-dimensional state space and a 30-dimensional action space, requiring high-dimensional control under contact-rich dynamics. As shown in Table 10, SAD-Flower is the only method that simultaneously satisfies safety and admissibility constraints while maintaining low dynamic-consistency violation. Other baselines either violate state or action constraints, or produce trajectories with larger dynamic inconsistency. These results further confirm that SAD-Flower scales to challenging dexterous manipulation settings while retaining reliable constraint enforcement.

**Effect of Injected Dynamics Noise** To further study the effect of dynamics-model error, we inject Gaussian noise into the learned Maze dynamics and evaluate SAD-Flower under increasing noise levels. This perturbation serves as a proxy for poor model conditions, such as discontinuities, unmodeled effects, or large prediction errors. As shown in Table 11, moderate noise is tolerated, while larger perturbations, corresponding to 10%–40% of a grid movement, lead to increasingly chattering trajectories and eventual constraint violations. These results support the observation that SAD-Flower is robust to moderate dynamics errors, but severe model mismatch can degrade dynamic consistency and constraint satisfaction.

**Flow Matching Implementation** We follow the planning setup of Diffuser (Janner et al., 2022) and train a flow matching planner together with a reward model using the same offline dataset. The flow matching model learns to generate full

*Table 12.* The Performance of the proposed SAD-Flower with QP and DPCC (Römer et al., 2025) with nonlinear optimization. Constraint for the Hopper is the same as the admissible heights in Table 2. Three DPCCs are proposed in their work. DPCC-T (Temporal consistency), which is shown in Table 3, selects the trajectory that deviates the least from the previous timestep, DPCC-C (Cumulative projection cost) selects the trajectory that has been modified the least by the projection operation, and DPCC-R (Random) selects the trajectory randomly.

| Experiment | Metric | DPCC-T | DPCC-C | DPCC-R | Ours |
|---|---|---|---|---|---|
| Hopper (Med-Expert) | safety | 0.01±0.03 | 0.02±0.02 | 0.02±0.02 | **0.00±0.00** |
| | admissib. | **0.00±0.00** | **0.00±0.00** | **0.00±0.00** | **0.00±0.00** |
| | dyn. consist. | **0.01±0.01** | **0.01±0.00** | **0.01±0.00** | **0.01±0.01** |
| | reward | 0.61±0.07 | 0.65±0.16 | 0.64±0.25 | **0.93±0.23** |
| | time | 4.24±1.64 | 4.56±1.72 | 6.21±2.61 | **0.06±0.00** |

state-action trajectories from a prior, while the reward model is used to evaluate the generated trajectories.

During inference, we use Monte Carlo selection, following the planning strategy studied in Feng et al. (2025). Specifically, we sample multiple unconditioned trajectory candidates from the pretrained flow matching model, evaluate each candidate using the learned reward model, and select the trajectory with the highest predicted return as the planned trajectory.

## D. Comparison with Constraint-Aware Baselines

We further compare SAD-Flower against two additional constraint-aware baselines: a reject-sampling method, which offers a simple heuristic for constraint handling, and CoBL-Diffusion (Mizuta & Leung, 2024), which leverages control-theoretic rewards to encourage constraint satisfaction.

The reject-sampling approach is a naive method for constraint satisfaction. It first trains a flow matching model on the original dataset. At inference, it samples a batch of trajectory candidates and discards those violating constraints. However, these candidates may violate test-time constraints since those constraints are unseen during training. As shown in Table 13, this approach consistently fails to satisfy constraints across all tasks, including Maze2d, Hopper, Walker2d, and Kuka Block-Stacking. Both safety and admissibility violations remain non-zero. This confirms that post-hoc filtering alone is insufficient when the generative model has no knowledge of constraint structure—especially under novel test-time constraints. Even though flow matching can model multimodal behavior patterns, it cannot reliably sample within unseen constraint sets unless such constraints are explicitly incorporated during inference.

CoBL-Diffusion injects physical constraints into a diffusion model by shaping its denoising process with auxiliary rewards derived from Control Barrier Functions (CBFs) and Control Lyapunov Functions (CLFs). This allows it to softly bias generation toward constraint-adherent behaviors, but without any formal guarantees. We evaluate CoBL-Diffusion in the LargeMaze environment, using the same horizon and constraints as in other baselines. As shown in Table 14, the method fails to fully enforce safety and admissibility constraints. This is expected given several key limitations: (1) it predicts only actions, whereas SAD-Flower jointly generates states and actions, which allows for more precise control of trajectory behavior; (2) its CBFs target only state constraints, while ours apply to both state and action spaces; (3) it uses CLFs for general stability, not for ensuring formal dynamic consistency as we do; and (4) it lacks prescribed-time scheduling, making it difficult to guarantee constraint satisfaction at the final step. These design choices limit CoBL-Diffusion's ability to robustly handle complex, temporally structured constraints. In contrast, SAD-Flower provides a framework to reliably enforce constraints even under novel test-time scenarios.

## E. Environment Details

We evaluate our method across a variety of trajectory planning domains, summarized in Table 15. The benchmark includes navigation (Maze2d-Umaze, Maze2d-Large), locomotion (Hopper, Walker2d), and robotic manipulation (Kuka Block-Stacking). Environments are simulated using MuJoCo (Todorov et al., 2012) or PyBullet (Coumans & Bai, 2016–2021), and cover a range of state and action dimensions.

We use offline datasets to train the generative and control models for each benchmark. As shown in Table 16, the datasets for locomotion and navigation tasks are retrieved from the D4RL benchmark suite (Fu et al., 2020), with varying sizes

*Table 13.* Performance of SAD-Flower and reject-sample method across all tasks.

| Experiment | Metric | Reject-sample | Ours |
|---|---|---|---|
| Maze2d (Large) | safety | $0.16 \pm 0.29$ | $\mathbf{0.00 \pm 0.00}$ |
| | admissib. | $0.11 \pm 0.00$ | $\mathbf{0.00 \pm 0.00}$ |
| | dyn. consist. | $0.02 \pm 0.01$ | $\mathbf{0.01 \pm 0.01}$ |
| | reward | $\mathbf{1.52 \pm 0.28}$ | $1.42 \pm 0.52$ |
| Maze2d (Umaze) | safety | $0.04 \pm 0.09$ | $\mathbf{0.00 \pm 0.00}$ |
| | admissib. | $0.10 \pm 0.01$ | $\mathbf{0.00 \pm 0.00}$ |
| | dyn. consist. | $\mathbf{0.01 \pm 0.01}$ | $\mathbf{0.01 \pm 0.01}$ |
| | reward | $\mathbf{2.91 \pm 0.65}$ | $2.66 \pm 0.88$ |
| Hopper (Med-Expert) | safety | $0.14 \pm 0.18$ | $\mathbf{0.00 \pm 0.00}$ |
| | admissib. | $0.05 \pm 0.02$ | $\mathbf{0.00 \pm 0.00}$ |
| | dyn. consist. | $0.02 \pm 0.01$ | $\mathbf{0.01 \pm 0.01}$ |
| | reward | $0.49 \pm 0.33$ | $\mathbf{0.93 \pm 0.23}$ |
| Hopper (Medium) | safety | $0.03 \pm 0.04$ | $\mathbf{0.00 \pm 0.00}$ |
| | admissib. | $0.06 \pm 0.03$ | $\mathbf{0.00 \pm 0.00}$ |
| | dyn. consist. | $0.02 \pm 0.02$ | $\mathbf{0.01 \pm 0.01}$ |
| | reward | $0.24 \pm 0.14$ | $\mathbf{0.34 \pm 0.03}$ |
| Walker2d (Med-Expert) | safety | $0.08 \pm 0.10$ | $\mathbf{0.00 \pm 0.00}$ |
| | admissib. | $0.08 \pm 0.08$ | $\mathbf{0.00 \pm 0.00}$ |
| | dyn. consist. | $\mathbf{0.04 \pm 0.06}$ | $\mathbf{0.04 \pm 0.04}$ |
| | reward | $\mathbf{0.97 \pm 0.22}$ | $0.89 \pm 0.32$ |
| Walker2d (Medium) | safety | $0.18 \pm 0.15$ | $\mathbf{0.00 \pm 0.00}$ |
| | admissib. | $0.16 \pm 0.10$ | $\mathbf{0.00 \pm 0.00}$ |
| | dyn. consist. | $\mathbf{0.07 \pm 0.17}$ | $\mathbf{0.07 \pm 0.15}$ |
| | reward | $\mathbf{0.68 \pm 0.19}$ | $0.42 \pm 0.23$ |
| KUKA Block Stacking | safety | $0.01 \pm 0.01$ | $\mathbf{0.00 \pm 0.00}$ |
| | reward | $0.44 \pm 0.67$ | $\mathbf{0.45 \pm 0.21}$ |

*Table 14.* Performance of SAD-Flower and CoBL-Diffusion in the navigation task.

| Experiment | Metric | CoBL-Diffusion | Ours |
|---|---|---|---|
| Maze2d (Large) | safety | $0.01 \pm 0.04$ | $\mathbf{0.00 \pm 0.00}$ |
| | admissib. | $0.23 \pm 0.33$ | $\mathbf{0.00 \pm 0.00}$ |
| | dyn. consist. | $\mathbf{0.00 \pm 0.00}$ | $0.01 \pm 0.01$ |
| | reward | $0.15 \pm 0.18$ | $\mathbf{1.42 \pm 0.52}$ |

(e.g., medium vs. medium-expert settings). For locomotion environments, datasets are collected using soft actor-critic (SAC) policies (Haarnoja et al., 2018), either partially trained or mixed with expert-level demonstrations. For the Kuka block-stacking task (Janner et al., 2022), data is generated via the PDDLStream planner (Garrett et al., 2020), which provides feasible robotic manipulation trajectories in structured stacking scenarios.

*Table 15.* Settings for each tasks.

| Environment | Simulator | Obs. Dim. | Action Dim. |
|---|---|---|---|
| Maze2d-Umaze-v1 | MuJoCo | 4 | 2 |
| Maze2d-Large-v1 | MuJoCo | 4 | 2 |
| Hopper | MuJoCo | 11 | 3 |
| Walker2d | MuJoCo | 17 | 6 |
| Kuka Block-Stacking | PyBullet | 39 | - |

*Table 16.* Dataset details for each benchmark environment, including the number of trajectories and the source or algorithm used to generate the data.

| Environment | # of Trajectories | Source / Generation Method |
|---|---|---|
| Maze2d-Umaze-v1 | $10^6$ | D4RL (Fu et al., 2020) |
| Maze2d-Large-v1 | $4 \times 10^6$ | D4RL (Fu et al., 2020) |
| Hopper-medium | $10^6$ | Partially trained SAC (Haarnoja et al., 2018) |
| Hopper-medium-expert | $2 \times 10^6$ | Mixture of expert and partial SAC |
| Walker2d-medium | $10^6$ | Partially trained SAC (Haarnoja et al., 2018) |
| Walker2d-medium-expert | $2 \times 10^6$ | Mixture of expert and partial SAC |
| Kuka Block-stacking | 10,000 | PDDLStream planner (Garrett et al., 2020) |

# F. Constraint Settings in Each Tasks

We introduce task-specific state and action constraints to evaluate the ability of generative planners to handle safety and admissibility under diverse and challenging conditions. The test-time constraints are deliberately chosen to be more restrictive and often unseen during training, making constraint satisfaction a non-trivial requirement.

**Maze2d.** The goal in Maze2d environments is to generate a feasible trajectory from a randomly sampled initial position to a randomly sampled goal location. The generative model is conditioned on both endpoints and produces full trajectories through the maze.

To evaluate constraint satisfaction, we introduce two novel, unseen obstacles at test time that do not completely block the path but add non-trivial planning constraints. The first is a superellipse-shaped obstacle defined as:

$$\left(\frac{x - x_0}{a}\right)^2 + \left(\frac{y - y_0}{b}\right)^2 \geq 1, \tag{14}$$

where $(x, y) \in \mathbb{R}^2$ is a trajectory state and $(x_0, y_0) \in \mathbb{R}^2$ is the center of the obstacle. The parameters $a > 0$ and $b > 0$ control the width and height of the obstacle.

The second is a higher-order polynomial barrier defined as:

$$\left(\frac{x - x_0}{a}\right)^4 + \left(\frac{y - y_0}{b}\right)^4 \geq 1. \tag{15}$$

This formulation results in sharper obstacle boundaries and makes naive post-processing or trajectory truncation ineffective due to the nonlinearity of the feasible region.

At test time, we tighten the original action constraint $\boldsymbol{a} \in [-1, 1]^2$ by a relative margin of 0.1 on each side, yielding

$$\boldsymbol{a} \in [-1 + 0.1, \, 1 - 0.1]^2 = [-0.9, 0.9]^2. \tag{16}$$

Both `Maze2d-Umaze-v1` and `Maze2d-Large-v1` contain these two novel obstacles at test time to assess generalization to unseen constraints.

**Hopper and Walker2d.** In the locomotion tasks, we impose test-time constraints that limit the robot's vertical motion to avoid collisions with overhead obstacles. Specifically, the height of the robot's torso must remain below a fixed roof height $z$, defined as:

$$s < z, \tag{17}$$

where $s$ is the vertical position of the torso. We evaluate this constraint under increasingly restrictive settings, such as $z = 1.6, 1.4$, and $1.25$, to assess robustness to constraint tightening.

This state constraint is often in conflict with the task objective, which rewards forward jumping or walking. As a result, satisfying the constraint typically requires sacrificing task performance. Additionally, the control inputs are constrained by an action box constraint:

$$\boldsymbol{a} \in [-1, 1]^d, \tag{18}$$

where $d$ is the dimensionality of the action space (3 for Hopper and 6 for Walker2d).

**Kuka Block-Stacking.** For the Kuka block-stacking task, the generative model is conditioned on object locations and outputs joint trajectories for manipulation. To simulate partial system degradation or workspace reconfiguration, we apply a tighter state constraint on the robot's joint positions. Specifically, we scale the original feasible joint limits by a factor of 0.9:

$$\boldsymbol{q} \in 0.9 \cdot [\boldsymbol{q}_{\min}, \boldsymbol{q}_{\max}], \tag{19}$$

where $\boldsymbol{q}_{\min}$ and $\boldsymbol{q}_{\max}$ represent the original lower and upper bounds of each joint. No explicit action constraint is applied in this task.

These constraints, particularly when unseen during training, pose a significant challenge for planning and provide a rigorous benchmark for evaluating constraint satisfaction and generalization capabilities.

## G. Training Details

We provide implementation details of all baseline methods and our proposed approach, including training configurations, hyperparameters, and model architectures.

All baseline methods are trained using their official or publicly available codebases: Diffuser follows the implementation of Janner et al. (2022), Truncation, Classifier Guidance, and SafeDiffuser are implemented based on Xiao et al. (2025), Decision Diffuser follows Ajay et al. (2023), and Flow Matching uses the codebase from Feng et al. (2025). Our method is built on top of the same Flow Matching framework from Feng et al. (2025).

**Hyperparameters.** Table 17 shows the training hyperparameters for all baselines and our method across different environments. All methods use the same horizon and batch size per environment. Note that our method shares training settings with the flow matching baseline to ensure a fair comparison.

**Forward Dynamics Model Training.** To support constraint enforcement in SAD-Flower, we train a forward dynamics model for each environment using the same dataset used to train the generative models. This ensures a fair comparison without introducing additional supervision. The training details are summarized in Table 18.

For the Maze2d environments, a known analytical dynamics model can also be derived:

$$\begin{bmatrix} x(k+1) \\ y(k+1) \\ v_x(k+1) \\ v_y(k+1) \end{bmatrix} = \begin{bmatrix} 1 & 0 & dt & 0 \\ 0 & 1 & 0 & dt \\ 0 & 0 & 1 & 0 \\ 0 & 0 & 0 & 1 \end{bmatrix} \begin{bmatrix} x(k) \\ y(k) \\ v_x(k) \\ v_y(k) \end{bmatrix} + \begin{bmatrix} 0.5\alpha dt^2 & 0 \\ 0 & 0.5\alpha dt^2 \\ \alpha dt & 0 \\ 0 & \alpha dt \end{bmatrix} \begin{bmatrix} u_x(k) \\ u_y(k) \end{bmatrix} \tag{20}$$

where $[x_k, y_k, v_{x,k}, v_{y,k}]^T$ is the system state representing position and velocity, $[u_{x,k}, u_{y,k}]^T$ is the input force, $dt$ is the simulation time step, and $\alpha$ is the gear ratio divided by mass (due to primitive joint control).

Table 17. Training hyperparameters for each method across environments.

| Maze2d-Umaze | Diffuser | Truncation | CG | S-Diffuser | FM | SAD-Flower |
|---|---|---|---|---|---|---|
| Batch Size | 32 | 32 | 32 | 32 | 32 | 32 |
| Learning Rate | $2e^{-4}$ | $2e^{-4}$ | $2e^{-4}$ | $2e^{-4}$ | $2e^{-4}$ | $2e^{-4}$ |
| Steps | $2e^6$ | $2e^6$ | $2e^6$ | $1e^6$ | $1e^6$ | $1e^6$ |
| **Maze2d-Large** | Diffuser | Truncation | CG | S-Diffuser | FM | SAD-Flower |
| Batch Size | 32 | 32 | 32 | 32 | 32 | 32 |
| Learning Rate | $2e^{-4}$ | $2e^{-4}$ | $2e^{-4}$ | $2e^{-4}$ | $2e^{-4}$ | $2e^{-4}$ |
| Steps | $2e^6$ | $2e^6$ | $2e^6$ | $1e^6$ | $1e^6$ | $1e^6$ |
| **Hopper** | Diffuser | Truncation | CG | S-Diffuser | FM | SAD-Flower |
| Batch Size | 32 | 32 | 32 | 32 | 32 | 32 |
| Learning Rate | $2e^{-4}$ | $2e^{-4}$ | $2e^{-4}$ | $2e^{-4}$ | $2e^{-4}$ | $2e^{-4}$ |
| Steps | $2e^6$ | $2e^6$ | $2e^6$ | $1e^6$ | $1e^6$ | $1e^6$ |
| **Walker2d** | Diffuser | Truncation | CG | S-Diffuser | FM | SAD-Flower |
| Batch Size | 32 | 32 | 32 | 32 | 32 | 32 |
| Learning Rate | $2e^{-4}$ | $2e^{-4}$ | $2e^{-4}$ | $2e^{-4}$ | $2e^{-4}$ | $2e^{-4}$ |
| Steps | $2e^6$ | $2e^6$ | $2e^6$ | $1e^6$ | $1e^6$ | $1e^6$ |
| **Kuka Block-Stacking** | Diffuser | Truncation | CG | S-Diffuser | FM | SAD-Flower |
| Batch Size | 32 | 32 | 32 | 32 | 32 | 32 |
| Learning Rate | $2e^{-5}$ | $2e^{-5}$ | $2e^{-5}$ | $2e^{-5}$ | $2e^{-4}$ | $2e^{-4}$ |
| Steps | $7e^5$ | $7e^5$ | $7e^5$ | $7e^5$ | $7e^5$ | $7e^5$ |

Table 18. Training settings for the forward dynamics models used in SAD-Flower.

| Environment | Batch Size | Learning Rate | Steps |
|---|---|---|---|
| Maze2d (Umaze/Large) | 256 | $1 \times 10^{-3}$ | $1 \times 10^6$ |
| Hopper | 256 | $1 \times 10^{-3}$ | $1 \times 10^6$ |
| Walker2d | 256 | $1 \times 10^{-3}$ | $1 \times 10^6$ |

**Model Architecture.** Diffusion-based models (Diffuser, Truncation, Classifier Guidance, SafeDiffuser) use the U-Net architecture with residual temporal convolutions, group normalization, and Mish nonlinearities, as described in (Janner et al., 2022). Flow Matching and SAD-Flower use the Transformer-based backbone proposed by (Feng et al., 2025), consisting of 8 layers with a hidden dimension of 256. The forward model in SAD-Flower is a feedforward neural network with 3 hidden layers of size 512, where the input dimension is the sum of the observation dimension and the action dimension, and the output is the observation dimension.

# H. Computational Resources

All experiments were conducted using four high-performance workstations with identical or near-identical configurations. Each workstation is equipped with an AMD EPYC series CPU, an NVIDIA Tesla P100 GPU, and 16 GB of GPU memory. The specific hardware details are summarized in Table 19.

Table 19. Hardware specifications of workstations used for training and evaluation.

| Workstation | CPU | GPU | GPU RAM |
|---|---|---|---|
| 1 | AMD EPYC 7542 | NVIDIA Tesla P100 | 16 GB |
| 2 | AMD EPYC 7542 | NVIDIA Tesla P100 | 16 GB |
| 3 | AMD EPYC 7742 | NVIDIA Tesla P100 | 16 GB |
| 4 | AMD EPYC 7542 | NVIDIA Tesla P100 | 16 GB |

## I. Computation Analysis

We report the detailed computational costs of SAD-Flower and all baseline methods across our benchmark tasks in Table 20. As expected, SAD-Flower incurs greater computational cost compared to unconstrained generative planners such as Diffuser and FM, due to the overhead introduced by enforcing multiple constraints during sampling. Nonetheless, in some domains—such as Hopper-Medium-Expert and Kuka Block-Stacking—SAD-Flower achieves a comparable runtime to SafeDiffuser, highlighting the efficiency gains enabled by our prescribed-time control formulation. Importantly, SAD-Flower is the only method that consistently satisfies all three constraint classes—state, action, and dynamic consistency—across all evaluated domains. No faster baseline provides this level of safety and reliability.

*Table 20.* Computation analysis (sec) of the proposed SAD-Flower and baselines across navigation, locomotion, and manipulation tasks.

| Experiment | Diffuser | Trunc | CG | FM | S-Diffuser | D-Diffuser | Ours |
|---|---|---|---|---|---|---|---|
| Maze2d (Large) | 0.01±0.01 | 0.02±0.01 | 0.04±0.01 | 0.06±0.02 | 0.06±0.01 | 0.01±0.01 | 0.28±0.02 |
| Maze2d (Umaze) | 0.01±0.01 | — | 0.04±0.01 | 0.02±0.01 | 0.06±0.02 | 0.01±0.01 | 0.09±0.01 |
| Hopper (Med-Expert) | 0.05±0.01 | 0.06±0.02 | 0.06±0.01 | 0.01±0.01 | 0.11±0.02 | 0.02±0.01 | 0.14±0.01 |
| Hopper (Medium) | 0.05±0.01 | 0.06±0.01 | 0.06±0.02 | 0.01±0.01 | 0.09±0.02 | 0.02±0.01 | 0.13±0.02 |
| Walker2D (Med-Expert) | 0.06±0.02 | 0.06±0.01 | 0.05±0.01 | 0.01±0.01 | 0.09±0.02 | 0.02±0.01 | 0.15±0.01 |
| Walker2D (Medium) | 0.04±0.01 | 0.06±0.03 | 0.07±0.02 | 0.01±0.01 | 0.10±0.02 | 0.02±0.01 | 0.14±0.01 |
| KUKA Block Stacking | 0.70±0.01 | 0.84±0.01 | 0.86±0.01 | 0.47±0.06 | 0.78±0.01 | 0.04±0.01 | 0.74±0.08 |

## J. Deployment on Real-World Robotic Platforms

Recent works have demonstrated the deployment of diffusion- and flow-based models on physical robotic systems (Chi et al., 2024; Yang et al., 2025), suggesting that generative trajectory models can be integrated into real-world pipelines involving perception, state estimation, and execution. SAD-Flower is designed as a planning-level method that generates a full trajectory over a finite horizon before execution. In this sense, it is open-loop within each rollout, but in practical robotic systems, such planners are typically used in a receding-horizon manner: plan, execute part of the trajectory, update the state estimate, and replan. This avoids strict real-time control-frequency requirements while still allowing feedback through replanning. During sampling, our method only adds a QP-based correction step to the generative process, so offline trajectory generation remains compatible with standard flow-based planning pipelines. Moreover, the resulting trajectories come with formal guarantees of safety, admissibility, and dynamic consistency under the assumptions of our theory.

At the same time, deploying SAD-Flower on physical robots introduces several practical challenges. Long-horizon planning and high-dimensional systems may increase computation due to repeated model inference and QP solving across denoising or flow-integration steps. Recent efficient generative planning methods, such as DiffuserLite (Dong et al., 2024) and Habi (Lu et al., 2025), may help reduce this cost and could be incorporated into our framework. In addition, SAD-Flower requires a dynamic model for enforcing dynamic consistency. Although such a model can be learned from onboard sensor data, sensor noise, model mismatch, and execution uncertainty must be carefully handled in real deployments. Uncertainty-aware models, such as Gaussian processes, and robust safety tools, such as robust CBFs, provide promising mechanisms for accounting for these effects. Another challenge is formulating safety constraints from high-dimensional observations, such as images or point clouds. However, this direction is compatible with our formulation: time-varying constraints, such as moving obstacles, can be handled using time-dependent CBFs, and the signed distance functions used in SAD-Flower provide a flexible obstacle representation (Long et al., 2021) that can be estimated from point clouds obtained by standard perception systems, such as LiDAR or depth cameras.

Overall, SAD-Flower provides a principled planning-level mechanism for generating constrained trajectories with formal guarantees, while introducing only a structured QP correction during sampling. Although real-robot deployment still requires addressing computation, model uncertainty, and perception-dependent constraint formulation, these challenges are active areas of research in robotics and generative planning. We believe they can be progressively addressed through improved sampling efficiency, robust dynamics learning, and better integration with perception modules, and we leave real-system deployment as an important direction for future work.

