# OpenReview forum: "SAD-Flower: Flow Matching for Safe, Admissible, and Dynamically Consistent Planning"
_ICML.cc/2026/Conference — ICML 2026 regular_

### Official Review · Reviewer_yRzA · 2026-03-11

**Soundness:** 3
**Presentation:** 4
**Significance:** 3
**Originality:** 3
**Overall Recommendation:** 5
**Confidence:** 3

**Summary:**

This paper introduces SAD-Flower, a control-augmented flow matching framework for trajectory planning that provides formal guarantees on safety, admissibility, and dynamic consistency. The key idea is to reinterpret the flow matching ODE as a controllable dynamical system by introducing a virtual control input. During sampling, the framework first follows the learned vector field without intervention (phase 1), then activates a constrained quadratic program (QP) controller that enforces Control Barrier Function (CBF) conditions for state/action constraints and a Control Lyapunov Function (CLF) condition for dynamic consistency (phase 2). Experiments on navigation, locomotion, and manipulation tasks demonstrate that SAD-Flower outperforms baselines.

**Compliance With Llm Reviewing Policy:**

Affirmed.

**Final Justification:**

I appreciate the comments during the rebuttal phase and I will maintain the positive recommendation.

**Key Questions For Authors:**

1. Theorem 5.1 guarantees constraint satisfaction if the QP is feasible for all t. In practice, how to guarantee the feasibility during further experiments in more complicated tasks? If infeasibility occurs, how to deal with it, or can you provide theoretical conditions (e.g., on the constraint sets or dynamics) that ensure feasibility for all trajectories?

2. The choice of T_0 appears critical (Fig. 3). How to select it for a new task? Is there a principled way to determine it, or does it require task-specific tuning?

3. Applications only on open-loop planning are quite limited. Are there any discussions about the gap between SAD-Flower and its advancement on closed-loop usage?

**Limitations:**

The authors adequately discuss their limitations in Impact Statement.

**Strengths And Weaknesses:**

**Strengths**

1. The paper is overall well written. The paper provides detailed theoretical construction of SAD-Flower and formal proofs (Theorems 5.1, B.1, B.2) that under QP feasibility, the generated trajectories satisfy state constraints, action constraints, and dynamic consistency at the final time.

2. Reformulating flow matching as a controllable system and embedding CBF/CLF conditions is a novel and technically sound synthesis of generative modeling and control theory.

3. Experiments cover diverse domains (navigation, locomotion, manipulation) with varying state/action dimensions. Robustness tests (stricter constraints, reduced data, imperfect dynamics) demonstrate practical reliability.

**Weaknesses**
1. While the paper claims feasibility "under mild conditions" and notes no infeasibility in experiments, this is not rigorously proven for arbitrary trajectories. Maybe this will limit the applications to more complicated planning scenarios and tasks?

2. The choice of when to activate control (T_0) significantly affects performance (shown in Fig. 3). The paper does not provide a principled method for selecting this activation time, which is treated as an empirical tuning parameter. This limits the framework's plug-and-play applicability across new tasks.

3. The experiments focus on relatively simple state/action constraints. It remains unclear how the method scales to complex, non-convex, or temporally extended constraints (e.g., collision avoidance with multiple dynamic obstacles). The CBF formulation relies on signed distance functions, which may be difficult to obtain for arbitrary environments.

4. The paper evaluates open-loop trajectory generation, but the ultimate goal is closed-loop planning. As mentioned in Appendix J, SAD-Flower is "designed as an open-loop planner", which remains the real-world application questionable.

---

> ### Author Rebuttal · Authors · 2026-03-30
>
> We thank the reviewer for the positive evaluation and appreciate the opportunity to address the questions.
>
> ## W1+Q1: QP feasibility
> ---
> We appreciate the reviewer’s question regarding feasibility. Ensuring feasibility for arbitrary initial states is inherently challenging, as there exist scenarios where no trajectory can simultaneously satisfy safety, admissibility, and dynamic consistency (e.g., a system moving at high speed toward an obstacle). Feasibility, therefore, depends jointly on the initial state, system dynamics, and constraints, making it inherently difficult to verify globally. In principle, feasibility could be verified via reachability analysis, but this is computationally prohibitive in practice, and any general sufficient condition is expected to be similarly expensive.
>
> Importantly, the complexity of the obstacle-free space is not causing difficulties in our approach by itself since our formulation mitigates this challenge by operating on integrator-like flow-matching dynamics (Remark 5.2), where feasibility is significantly less restrictive for simple dynamics, e.g., point mass, even in complex environments. In addition, Lemmas B.1 and B.2 show that the individual CBF and CLF constraints are feasible under mild conditions for our specific construction. Thus, while global feasibility cannot be guaranteed in general, the structure of our formulation makes infeasibility much less likely in practice.
>
> Quantifying “closeness to infeasibility” would require estimating the volume of the feasible set, which is intractable for the problems we consider. We therefore report empirical violations instead. Across all experiments, safety and admissibility violations are absent, and dynamic-consistency errors remain small.
>
> Finally, if feasibility would be lost, one could always resort to slack-variable relaxations as discussed in Remark 5.2, which are commonly used in many practically employed approaches targeted at maintaining safety.
>
> ## W2+Q2: Selecting the control activation time T_0
> ---
> We thank the reviewer for highlighting this point. We would like to clarify that T_0 does not critically affect performance across its full range. As shown in Fig. 3, sensitivity is limited to the early regime (T_0≤0.3), where activating control too early interferes with the learned generative dynamics. This effect has also been observed in prior work [1].
>
> Beyond this region, performance and dynamic consistency remain stable, with a broad plateau for T_0≳0.4. In practice, this indicates that T_0 does not require fine-tuning.
>
> A more principled alternative would be to activate control based on a constraint-violation indicator during sampling rather than a fixed threshold, which we consider an interesting direction for future work.
>
> ## W3: Scalability to complex and non-convex constraints
> ---
> We thank the reviewer for raising this question. Our method does not require convex constraints. The CBF formulation operates directly on state space via the flow-matching dynamics, allowing it to handle complex and non-convex environments. We refer to our discussion in W1 for additional details on scalability.
>
> Time-varying constraints (e.g., moving obstacles) can be incorporated by extending the CBF to a time-dependent formulation, which is well established in the literature. Additionally, signed distance functions, which used in our formulation, provide a flexible and practical representation of obstacles [2], and can be obtained from point cloud data obtained via standard perception systems (e.g., LiDAR or depth cameras).
>
> ## W4+Q3: Open-loop vs. closed-loop applicability
> ---
> We appreciate the reviewer’s question regarding practical deployment. We clarify that “open-loop” in our paper refers to generating and executing a trajectory over a finite horizon within a single rollout. In practice, this is typically combined with replanning in a manner (plan → execute → replan), which is an effective strategy in robotic systems.
>
> Under this interpretation, SAD-Flower can be directly used in a closed-loop setting in a receding-horizon manner, as commonly done in generative planners (e.g., Diffuser).
>
> We note that extending to fully closed-loop guarantees introduces additional challenges, such as recursive feasibility and stability across replanning steps, which are generally not addressed in current generative planning frameworks. While our method does not explicitly resolve these challenges, it can be integrated into receding-horizon or MPC-style control loops, and we view this as an important direction for future work.
>
> *[1] Fan, W., et al. Cfg-zero⋆: Improved classifier-free guidance for flow matching models. arXiv preprint arXiv:2503.18886, 2025.*
>
> *[2] Long, K., et al. Learning barrier functions with memory for robust safe navigation. IEEE Robotics and Automation Letters, 2021.*

---

> > ### Author Rebuttal · Reviewer_yRzA · 2026-04-01
> >
> > The authors address my concern and thus I maintain my positive score.

---

> > > ### Author Response · Authors · 2026-04-02
> > >
> > > We appreciate your insightful comments and positive score.

---

### Official Review · Reviewer_vgKW · 2026-03-12

**Soundness:** 3
**Presentation:** 3
**Significance:** 3
**Originality:** 3
**Overall Recommendation:** 4
**Confidence:** 4

**Summary:**

This paper proposes SAD-Flower, a control-augmented flow matching framework for constrained planning. The core problem addressed is that existing generative planners often produce plausible-looking trajectories that nevertheless fail to satisfy the constraints required for real-world execution. The authors argue that practical deployment demands the simultaneous satisfaction of three key properties: Safety (state constraints), Executability (action constraints), and Dynamical Consistency (generated trajectories must obey the system dynamics).

To this end, the authors reformulate the generative process as a controllable dynamical system with virtual control inputs. During sampling, tools from control theory—especially CBF/CLF-type constraints enforced via quadratic programming (QP)—are used to guarantee the desired properties. The claimed advantage is that the model does not require retraining when new constraints are introduced. Formal results show that constraint satisfaction can be ensured under the condition that the QP is feasible. Experimental validation is carried out on Maze2d, Hopper, Walker2d, and the Kuka block-stacking task.

**Compliance With Llm Reviewing Policy:**

Affirmed.

**Key Questions For Authors:**

1. How close to infeasibility does the sampling-time QP get across the evaluated tasks, and are there representative failure cases where feasibility becomes numerically delicate?
2. Can the authors provide a more explicit breakdown of total runtime into base flow-matching integration versus QP overhead, ideally as a function of horizon or dimension?
3. The imperfect-dynamics study is useful, but how severe can model misspecification become before the practical safety advantage materially degrades?
4. Ablation of control ingredients: How much of the observed gain is attributable specifically to dynamic-consistency enforcement, as opposed to only the safety and admissibility constraints?

**Limitations:**

1. The theorem depends on QP feasibility. This is a natural assumption, but it is still a condition rather than an unconditional guarantee.
2. The method depends on a forward model. When the forward model is learned, practical guarantees inevitably weaken relative to the idealized analysis.
3. Computational cost may grow with horizon and dimension. The paper argues the added QP is efficient, but more detailed scaling evidence would help.

**Strengths And Weaknesses:**

Strengths
1. Generative planners are attractive, but without executable constraint satisfaction they are hard to trust in robotics or safety-critical planning. This paper squarely targets that gap.
2. It does not stop at state-safety constraints; it also addresses action feasibility and dynamic consistency, which are crucial in practice.
3. The QP-based control augmentation is not an afterthought; it is central to the method and tightly connected to the theoretical guarantees.

Weaknesses
1. The strongest guarantees are conditional. The main theorem assumes QP feasibility, and in practice feasibility depends on the geometry of the constraints and the quality of the learned dynamics model.
2. Dynamic consistency is only approximately achieved in experiments. The paper explains that residual violations are small and mainly due to numerical integration, but this still matters when interpreting the formal claims.
3. Although the paper discusses robustness under imperfect dynamics, the method still inherits a nontrivial dependence on model quality.
4. The practical guarantees differ from the theorem-level guarantees. The paper could be even more explicit in separating “provable in the idealized formulation” from “empirically observed with learned models and numerical solvers.”

---

> ### Author Rebuttal · Authors · 2026-03-30
>
> We appreciate the reviewer’s positive feedback and respond to the questions below.
>
> ## W1+Q1: QP feasibility and failure case
> ---
> We appreciate the reviewer’s question regarding feasibility. QP feasibility is a common conditional assumption in control-based formulations. In our setting, feasibility is less restrictive than in classical control, since CBF/CLF constraints are imposed on the integrator-like flow-matching dynamics rather than the original system dynamics. In addition, Lemmas B.1–B.2 show that individual CBF and CLF constraints are feasible under mild conditions. Empirically, we did not observe infeasible or numerically delicate QPs across our experiments.
>
> Quantifying “closeness to infeasibility” is generally intractable, as it would require estimating the volume of the feasible set. In practice, we therefore report constraint violations, which are consistently absent for safety and admissibility, and minor for dynamic consistency.
>
> We refer to our response to W1+Q1 of reviewer yRzA for a more detailed discussion.
>
> ## Q2: Runtime breakdown with horizon and system dimension
> ---
> We thank the reviewer for requesting a more explicit breakdown. Appendix I (Table 17, columns “FM” vs. “Ours”) already separates the runtime of the base flow-matching integration from the additional QP overhead, allowing a direct comparison of their contributions.
>
> To further clarify scaling behavior, we conduct additional ablations: (i) varying the horizon (2, 8, 32, 128) on UMaze with fixed state dimension, and (ii) varying the system dimension (LargeMaze, Hopper, Walker2d) with fixed horizon. The [Table D and E (click here)](https://github.com/sadflowerplanning/additional_fig_table/blob/main/comptime/ablation_comp_time.md ) show that runtime increases mildly for short horizons and approximately linearly beyond H>32, while scaling linearly with state/action dimensionality.
>
> Together, these results provide a clear breakdown of how runtime decomposes and scales with both horizon length and system dimension.
>
> ## W3: Sensitivity to severe model misspecification
> ---
> We thank the reviewer for highlighting this point. As discussed in our response to Q1 from reviewer nsKa (Table A.), SAD-Flower remains robust under moderate model errors (e.g., down to 1% training data), with degradation appearing only under severe misspecification. Consistent behavior is observed in our noise-injection study, where large perturbations lead to violations. This indicates that safety degrades only under substantial model errors, while remaining stable in practical regimes. We refer the reviewer to the response to Q1 from reviewer nsKa for additional details.
>
> ## W2+W4: Gap between theoretical guarantees and practical behavior
> ---
> We agree with the reviewer that a clearer distinction between the idealized theoretical formulation and the empirically observed behavior would improve the presentation. We will incorporate this clarification into the abstract, introduction, and experimental sections.
>
> Regarding the small dynamic-consistency residuals, the theoretical guarantee applies to the continuous-time controlled ODE, for which the prescribed-time CLF ensures zero violation. In practice, however, this system cannot be solved exactly and must be approximated using numerical solvers. These solvers evaluate the control law at discrete time steps, effectively resulting in a sampled-data (zero-order hold) implementation. This introduces discretization errors in the Lyapunov analysis, which can explain the small residual violations observed in dynamic consistency.
>
> In principle, these errors can be reduced by increasing the integration accuracy (e.g., smaller step sizes), at the cost of additional computation. We believe that explicitly separating the idealized setting from the practical implementation will help clarify this point. From a practical perspective, perfect dynamic consistency is also not essential, similarly to the case with imperfect system models discussed in Remark 5.3.
>
> ## Q4: Ablation of control ingredients:
> ---
> We appreciate the reviewer’s interest in disentangling the contributions of different control components. Removing the CLF (i.e., enforcing only safety/admissibility via CBF) leads to dynamic inconsistency, where the planned actions do not realize the planned state trajectory. In practice, this can result in unsafe behavior and known issues such as the local trap phenomenon observed in prior work [1].
>
> We verify this through an ablation where only CBF constraints are applied. The [Figure A (click here)](https://github.com/sadflowerplanning/additional_fig_table/blob/main/No_CLF/affect_noCLF.md ) shows that the realized trajectories can drift into unsafe regions, and the planned trajectory exhibits local trapping behavior, confirming that dynamic-consistency enforcement via CLF is essential for reliable planning.
>
> *[1] Xiao, W., et al. Safediffuser: Safe planning with diffusion probabilistic models. ICLR, 2025.*

---

> > ### Author Rebuttal · Reviewer_vgKW · 2026-04-03
> >
> > The rebuttal satisfactorily addressed my main concerns.

---

> > > ### Author Response · Authors · 2026-04-04
> > >
> > > Thank you for considering our rebuttal. We appreciate the effort and time you put into evaluating our work.

---

### Official Review · Reviewer_eTef · 2026-03-13

**Soundness:** 3
**Presentation:** 2
**Significance:** 2
**Originality:** 3
**Overall Recommendation:** 4
**Confidence:** 3

**Summary:**

This paper proposes a control-augmented flow matching framework that enforces safety, admissibility, and dynamic consistency in trajectory planning. It imposes constraints through a quadratic program without retraining and provides formal guarantees for test-time constraint satisfaction. Experimental results show that the proposed method outperforms existing approaches on various robotic tasks.

**Compliance With Llm Reviewing Policy:**

Affirmed.

**Final Justification:**

The rebuttal has addressed my main concerns.

**Key Questions For Authors:**

1. Could the authors provide a more comprehensive comparison with other conditional sampling and sampling-then-verification methods, especially regarding flexibility, efficiency, and behavior under complex or degenerate constraints?
2. Is the proposed SAD-Flower framework generalizable to other generative planners like diffusion models? If so, what modifications are needed, and will the theoretical guarantees still hold?
3. How do the authors plan to reduce the inference overhead introduced by quadratic programming to enable efficient deployment in real-time or online replanning systems?
4. Are there any failure cases that the proposed method fail to ensure safety, admissibility, and dynamic consistency?

**Limitations:**

Yes

**Strengths And Weaknesses:**

Strength:
1. This work addresses a critical and practical issue in generative trajectory planning, i.e., lack of formal guarantees for safety, admissibility, and dynamic consistency. It integrates Control Barrier Functions and Control Lyapunov Functions into flow matching, providing solid theoretical guarantees.
2. The framework is well-designed with a two-stage generation pipeline and prescribed-time control, balancing trajectory diversity and constraint robustness. The proposed method enforces test-time constraints without retraining.
3. The experiments are extensive and comprehensive, demonstrating the effectiveness of the proposed method.

Weakness:
1. The proposed method can be viewed as a conditional sampling approach with strong guarantees. However, the authors do not systematically compare with other conditional sampling methods or sampling-then-verification techniques in terms of flexibility, efficiency, and failure modes under complex constraints.
2. The proposed method is tightly integrated with flow matching and does not explore its generalizability to other popular generative planners such as diffusion models. It remains unclear whether the control-theoretic formulation and theoretical guarantees can be easily extended or retain similar benefits in other generative sampling paradigms.
3. The additional quadratic programming increases inference overhead, which may restrict deployment in highly time-critical online replanning systems.

---

> ### Author Rebuttal · Authors · 2026-03-30
>
> We thank the reviewer for the feedback and are happy to reply to the questions.
>
> ## W1+Q1: Comparison with conditional sampling methods or sampling-then-verification techniques
> ---
> We appreciate the reviewer’s suggestion to clarify this comparison. To ensure alignment, our understanding is that conditional sampling methods incorporate constraint information during generation (e.g., via guidance or conditioning), while sampling-then-verification methods generate candidate trajectories and subsequently filter or select feasible ones. Under this interpretation, the current evaluation in our paper already covers both categories.
>
> Specifically, Table 2 includes **CG**, **D-Diffuser**, **Trunc**, and **S-Diffuser**, which represent different forms of conditional or guided sampling. Append. D / Table 10 additionally includes **Reject sampling**, which is a direct sampling-then-verification baseline. Across these methods, we observe complementary failure modes: guided methods improve some constraints but do not jointly ensure safety, admissibility, and dynamic consistency, while reject sampling fails under unseen test-time constraints because feasible trajectories have very low probability under the unconstrained model.
>
> We agree that making this categorization more explicit would improve clarity, and will revise the paper accordingly.
> ## W2+Q2: Generalization to other generative planners, e.g., diffusion models:
> ---
>
> Our control-theoretic formulation is not inherently limited to flow matching and can be extended to other generative planners such as diffusion models. We focus on flow matching because it already provides a general ODE-based formulation of generative sampling, encompassing a broad class of probability paths beyond the Gaussian-noise trajectories used in diffusion [1]. As a result, evaluating on flow matching already captures the setting relevant to ODE-based diffusion samplers.
>
> From a technical perspective, flow matching yields a deterministic continuous-time ODE [1], enabling direct use of standard CBF/CLF tools. Diffusion models are typically formulated as SDEs [2], where extensions would require stochastic CBF/CLF formulations. However, when diffusion models are converted to ODE-based samplers [2], our framework, e.g., formula (4), (5) can be applied in the same way.
>
> Overall, SAD-Flower can be viewed as a control layer for ODE-based generative sampling, and flow matching provides a sufficiently general setting to demonstrate this without requiring separate experiments on diffusion models.
>
> ## W3+Q3: Reducing QP overhead for real-time deployment
> ---
>
> We appreciate the reviewer’s concern regarding computational overhead. As with most approaches that enforce constraints at inference time, additional computation is required; however, our QP-based formulation is already efficient in practice, as shown in Table 3, where it compares favorably to related baselines.
>
> Importantly, our formulation admits several avenues for further acceleration. In particular, QPs can admit closed-form solutions [3], and recent work explores combining multiple CBF constraints into a single non-smooth CBF [4], which could lead to simplified expressions with negligible runtime overhead. While such solutions are not yet plug-and-play for general settings, this is an active research direction in the control community, and advances in this area can be directly integrated into our framework.
>
> We will revise the paper to include a clearer discussion of these opportunities.
>
> ## Q4: Failure cases and limitations of constraint guarantees
> ---
>
> We thank the reviewer for raising this point. In our experiments, we do not observe violations of safety or admissibility, and dynamic-consistency errors remain small.
>
> Failures may occur in scenarios where no feasible trajectory exists, or more generally when the underlying assumptions (e.g., model accuracy or constraint compatibility) are significantly violated. Such situations are inherent to constrained planning problems rather than specific to our method, and can be difficult to fully characterize in general.
>
> In our setting, feasibility is less restrictive than in classical control, since constraints are imposed on integrator-like flow-matching dynamics. Moreover, individual CBF and CLF constraints are feasible under mild conditions (Lemmas B.1–B.2), which further reduces the likelihood of infeasibility in practice. Empirically, we do not observe infeasibility issues empirically.
>
> We refer to our response to W1+Q1 of reviewer yRzA for a more detailed discussion.
>
> *[1] Lipman, Y., et al. Flow matching for generative modeling. ICLR, 2023.*
>
> *[2] Song, Y., et al. Score-based generative modeling through stochastic differential equations. ICLR, 2021.*
>
> *[3] Ong, P., et al. Universal formula for smooth safe stabilization. IEEE CDC, 2019.*
>
> *[4] Glotfelter, P., et al. Nonsmooth barrier functions with applications to multi-robot systems. IEEE Control Systems Letters, 2017.*

---

> > ### Author Rebuttal · Reviewer_eTef · 2026-04-02
> >
> > The responses have addressed all of my concerns. Thanks!

---

> > > ### Author Response · Authors · 2026-04-02
> > >
> > > Thank you for your comments and for raising your score. We appreciate the time and effort you put into evaluating our work!

---

### Official Review · Reviewer_nsKa · 2026-03-16

**Soundness:** 4
**Presentation:** 4
**Significance:** 3
**Originality:** 3
**Overall Recommendation:** 5
**Confidence:** 4

**Summary:**

The paper tackles a limitation of generative trajectory planners: standard flow-matching (FM) and diffusion planners can generate trajectories that look plausible but may violate safety constraints, action limits, or the underlying system dynamics. The paper aims to address this challenge by proposing SAD-Flower, a method that augments flow matching at test time with a virtual control input and then uses control barrier functions (CBFs) for state/action constraints and a control Lyapunov function (CLF) for dynamic consistency. These are enforced through a quadratic program during sampling, so the method does not require retraining when constraints change at test time.

The proposed method turns the FM sampling ODE into a controlled dynamical system. It leaves the early part of sampling uncontrolled to preserve sample quality, then activates constrained control later in the trajectory generation process. The paper claims that, under QP feasibility, the final sampled trajectory satisfies safety, admissibility, and dynamic consistency at the terminal time.

The method is evaluated on Maze2D, Hopper, Walker2d, Block Stacking with a robot arm, and a higher-dimensional Adroit Relocate setting. Across the reported results, SAD-Flower usually achieves zero safety and admissibility violations and very small dynamic consistency error, while keeping reward competitive with baselines.

**Compliance With Llm Reviewing Policy:**

Affirmed.

**Final Justification:**

The rebuttal addressed my main concerns.

**Key Questions For Authors:**

1. Dynamic consistency depends on access to the dynamics model. The paper does acknowledge this and studies robustness to imperfect models, but this still means reliability may degrade in contact-rich or poorly-modeled settings. Can you provide more experiments in contact-rich environments where the model is hard to learn?

2. The evaluation is good, but could be improved. For example, Block Stacking only tests state feasibility, not the full SAD-Flower setting, and the Adroit experiment compares only against vanilla FM rather than a broader set of stronger constrained baselines. Can you include these experiments?

**Limitations:**

Yes

**Strengths And Weaknesses:**

One of the main strengths of the paper is the principled integration of control theory and generative modelling. Recasting flow matching as a controlled system and then enforcing CBF/CLF conditions via a QP is a good idea. The paper also provides an explicit theorem stating terminal satisfaction of the three requirements under feasibility.

The empirical results are also strong. In Table 2, SAD-Flower is the only method that consistently shows no safety and admissibility violation across the main tasks while keeping rewards in the same ballpark as strong baselines; in Hopper and Walker2d it also substantially improves dynamic consistency relative to most baselines.

I don't see any major weaknesses in terms of the theoretical formulation or experiments of the paper.

---

> ### Author Rebuttal · Authors · 2026-03-30
>
> We thank the reviewer for the encouraging assessment and address the questions as follows.
>
>
> ## Q1: Performance degradation with poor models
> ---
> We appreciate the reviewer’s comments. In our paper, we already include a contact-rich benchmark (dexterous grasping task), demonstrating that SAD-Flower remains effective in high-dimensional dexterous manipulation.
>
> To further study the impact of imperfect dynamics, particularly relevant in contact-rich settings where accurate modeling is difficult, we extend our analysis by training the forward model on progressively smaller datasets. As shown in Table A, we observe that the method remains robust down to 1% of the data, while more severe degradation (e.g., 0.1%, 0.01%, where the data numbers are merely 1000 and 100) leads to safety violations. This indicates that SAD-Flower is tolerant to moderate model inaccuracies, but can degrade when the model becomes highly inaccurate.
>
> *Table A: Effect of training dataset size on dynamic consistency and constraint satisfaction (Hopper-Medium).*
> | Dataset ratio| Data number| Safety ($\downarrow$) | Admissibility ($\downarrow$) | Dynamic consistency ($\downarrow$) | score ($\uparrow$) |
> |--------|------|------------------------|-------------------------------|------------------------------------|---------------------|
> | $100\%$   | $10^6$ | $0.00 \pm 0.00$        | $0.00 \pm 0.00$               | $0.01 \pm 0.01$                    | $0.34 \pm 0.01$     |
> | $10\%$    | $10^5$ | $0.00 \pm 0.00$       | $0.00 \pm 0.00$               | $0.01 \pm 0.01$                    | $0.38 \pm 0.03$     |
> | $1\%$     | $10^4$ | $0.00 \pm 0.00$       | $0.00 \pm 0.00$               | $0.01 \pm 0.01$                    | $0.34 \pm 0.03$     |
> | $0.1\%$   | $10^3$ | $0.07 \pm 0.03$       | $0.00 \pm 0.00$               | $0.02 \pm 0.01$                    | $0.37 \pm 0.04$     |
> | $0.01\%$  | $10^2$  | $0.02 \pm 0.09$       | $0.00 \pm 0.00$               | $0.04 \pm 0.02$                    | $0.36 \pm 0.03$     |
>
> We further validate this trend by injecting Gaussian noise into the learned dynamics in [Table C (click here)](https://github.com/sadflowerplanning/additional_fig_table/blob/main/inject_noise_maze/inject_noise_maze.md). This can be viewed as a proxy for poor models in contact-rich environments, where discontinuities and unmodeled contacts introduce significant prediction errors. The noise levels are substantial (e.g., 10%–40% of a grid movement in Maze), and we observe that larger noise leads to chatter trajectories and constraint violations. This supports the observation that poor model quality, e.g., in a contact-rich setting, can affect performance, particularly through dynamic-consistency errors. These results help to provide a clear characterization of how model accuracy impacts performance.
>
> ## Q2: Improving the evaluation
> ---
> We thank the reviewer for the suggestion to strengthen the evaluation. For the KUKA block stacking task, our design choice is to evaluate whether SAD-Flower remains effective when only part of the full framework is needed. This benchmark (introduced in Diffuser) only exposes state constraints, with control handled by an external controller. We therefore include it to show that SAD-Flower is not restricted to enforcing all constraints jointly, but also performs well when only safety constraints are relevant.
>
> For the Adroit(dexterous grasping task) experiment, we include all baselines as shown in Table B. The results show that SAD-Flower is the only method that simultaneously achieves safety, admissibility, and strong dynamic consistency, while maintaining competitive task performance.
>
> We will further refine the presentation of these comparisons in the revised paper to make this more explicit.
>
> *Table B. Performance of all baselines and our method in a dexterous grasping scenario (Adroit-Hand for Relocate tasks).*
>
> | Methods | Safety ($\downarrow$) | Admissibility ($\downarrow$) | Dynamic consistency ($\downarrow$) | score ($\uparrow$) |
> |---------|--------|-----------|----------------|--------|
> | Diffuser | 2.53±0.78 | 0.32±0.07 | 0.09±0.04 | 1.04±0.07 |
> | Classifier Guidance | 2.51±0.77 | 0.31±0.07 | 0.15±0.06 | 1.03±0.06 |
> | Truncation | **0.00±0.00** | 0.32±0.07 | 0.11±0.03 | 1.04±0.06 |
> | SafeDiffuser | **0.00±0.00** | 0.30±0.06 | 0.13±0.05 | 1.01±0.06 |
> | DecisionDiffuser | 2.21±0.47 | 0.23±0.06 | 0.23±0.08 | 1.06±0.07 |
> | Flow Matching | 0.15±0.21 | 0.62±0.19 | 0.07±0.04 | **1.07±0.08** |
> | **Ours** | **0.00±0.00** | **0.00±0.00** | **0.06±0.09** | 1.05±0.23 |

---

> > ### Author Rebuttal · Reviewer_nsKa · 2026-04-03
> >
> > I thank the authors for their responses and for the additional experiments. I have no additional concerns.

---

> > > ### Author Response · Authors · 2026-04-03
> > >
> > > We appreciate your comments and raising the score! Thanks for the time and effort you put into our work.

---

### Decision · Program_Chairs · 2026-04-30

**Decision:**

Accept (regular)

**Comment:**

The paper proposes SAD-Flower, a control-augmented flow matching framework that enforces safety, admissibility, and dynamic consistency via CBF/CLF constraints solved using Quadratic Programming.
Novelty:  Recasting flow matching as a controllable dynamical system and integrating control-theoretic tools (CBFs/CLFs) into generative sampling is a principled and non-trivial contribution. The framework cleanly bridges generative modeling and control, and the ability to enforce constraints at test time without retraining is particularly appealing. Overall, a very interesting and elegant idea.
Evaluation: The empirical results are comprehensive and convincing, spanning multiple domains (navigation, locomotion, manipulation) and showing clear gains in constraint satisfaction while maintaining competitive reward.  However, comparisons to other conditional sampling or constraint-enforcement approaches are limited, making it harder to position the method relative to alternative pipelines. There are also some practical gaps, such as sensitivity to design choices (e.g., control activation time) and lack of evaluation in closed-loop settings.